# Nrm1 is a bistable switch connecting cell cycle progression to transcriptional control

Guillem Murciano-Julià [iD][1], Montserrat Vega [iD][1], Esther Pazo[1], Àlex Pascual-Serra[2], Isabel Alves-Rodrigues[1], Oriol Bagudanch[1], Roger Anglada[3], Núria Bonet [iD][3], Rosa Aligué[2], Sergio Moreno [iD][4], Baldo Oliva[5], Elena Hidalgo [iD][1] & José Ayté [iD][1✉]

## Abstract

Entry into the cell cycle requires activation of G1 cyclin-dependent kinases (CDKs) and the G1/S transcriptional program. In fission yeast, the MBF complex is the main transcription factor driving early cell-cycle gene expression. MBF-dependent transcription is activated in metaphase and repressed at the end of S phase by a feedback loop involving the cyclin Cig2 and co-repressors Nrm1 and Yox1. While replicative stress inactivates Yox1 via phosphorylation, the mechanism that activates MBF during an unperturbed cell cycle remains unclear. Here, we identify Nrm1 as the key target of cell cycle regulation in a two-step control mechanism. First, CDK1 phosphorylates Nrm1 in metaphase, leading to its release—along with Yox1—from chromatin. Second, unphosphorylated Nrm1, generated either by dephosphorylation or de novo synthesis, is degraded during anaphase, preventing its re-association with MBF until the end of the next S phase. Together, these parallel pathways create a precisely timed window of MBF activation, ensuring proper cell cycle progression and preserving genomic stability.

Keywords APC/C; CDK1; MBF; Nrm1; START
Subject Category Cell Cycle

## Introduction

The G1 phase is a pivotal stage in the eukaryotic cell cycle, including a key decision point called Start, which is also known as the Restriction Point in mammalian cells. This decision point occurs in late G1 and determines whether cells will proceed with proliferation in the vegetative cycle, remain in G1 to enter the sexual cycle, or transition into a quiescent state. Passage through Start irreversibly commits cells to the subsequent mitotic cycle, leading to chromosome replication during S phase. If regulation of this restriction point is lost, misregulated cell proliferation can occur, which is a hallmark of cancer (Pardee, 1989). In fission yeast,

passage through Start involves two essential steps: activation of the G1 cyclin-dependent kinase (CDK) and induction of the G1/S transcriptional program. Two critical regulators for this process have been identified: Cdc2/CDK1, the single CDK in fission yeast, and Cdc10. The precise role of Cdc2 at Start remains incompletely defined, other than that it is essential for cells to pass through Start; on the other hand, Cdc10 is a core component of the G1/S transcription factor MBF (Simanis et al, 1987). However, whether these activities are linked at Start remains unclear, with some studies suggesting that CDK is required to activate MBF, while others report that MBF activation is independent of CDK (Baum et al, 1998; Baum et al, 1997; Connolly et al, 1997; Reymond et al, 1993).

The MBF complex is large and includes at least three core proteins [Cdc10, Res1, and Res2 (Ayte et al, 1995)]; one co-activator [Rep2 during the mitotic cycle or Rep1 during meiosis (Nakashima et al, 1995; Sugiyama et al, 1994)]; and two co-repressors [Yox1 and Nrm1 (de Bruin et al, 2008; Gómez-Escoda et al, 2011)], which mainly associate with MBF during the G2 phase of the cell cycle. Similar to its metazoan functional counterpart, the E2F/pRB complex, MBF requires strict regulation to ensure proper cell cycle progression. For instance, cells with hypoactive MBF fail to complete S phase, leading to replicative stress (Tanaka et al, 1992), while hyperactive MBF can cause genomic instability (Caetano et al, 2014). To prevent such dysregulation, MBF-dependent transcription is repressed at the end of S phase by inactivation of the complex through a robust negative feedback loop. This loop involves the binding of the co-repressors Nrm1 and Yox1 to the MBF core protein, Cdc10 (de Bruin et al, 2008; Gómez-Escoda et al, 2011). This repression is further strengthened by CDK1/Cig2-mediated phosphorylation of Res1, the DNA-binding subunit of MBF (Ayté et al, 2001). Importantly, the key components of this negative feedback loop -including *cig2*, *yox1*, and *nrm1*- are regulated by MBF transcriptional activity (Ayté et al, 2001; de Bruin et al, 2008; Gómez-Escoda et al, 2011). This regulatory network ensures that MBF-dependent transcription is down-regulated as cells transition from S phase to G2 phase.

The regulation of the G1-to-S transcriptional wave by cell cycle checkpoints is highly conserved across species, from yeasts to higher eukaryotes. For example, under replicative stress,

[1]Oxidative Stress and Cell Cycle Group, Universitat Pompeu Fabra, Barcelona, Spain. [2]Department of Biomedical Science, University of Barcelona, Barcelona, Spain. [3]Genomics Core Facility, Universitat Pompeu Fabra, Barcelona, Spain. [4]Instituto de Biología Funcional y Genómica, CSIC, University of Salamanca, Salamanca, Spain. [5]Structural Bioinformatics Lab (GRIB), Universitat Pompeu Fabra, Barcelona, Spain. ✉E-mail: jose.ayte@upf.edu

checkpoints result in the constitutive activation of MBF in yeast (Caetano et al, 2011; Dutta et al, 2008; Gómez-Escoda et al, 2011) or E2F in metazoans (Bertoli et al, 2013a). Similarly, the DNA damage checkpoint inactivates the G1/S transcriptional program in both yeast (Ivanova et al, 2013) and mammalian cells (Inoue et al, 2007; Stevens et al, 2003; Zalmas et al, 2008). However, the mechanisms underlying MBF activation at the beginning of an unperturbed cell cycle remain unknown. The MBF co-activator Rep2 has been shown to associate with the complex, although its exact role in the regulation of MBF during the cell cycle is not fully understood (Chu et al, 2009; Eshaghi et al, 2011). In this study, we investigate the molecular mechanisms involved in MBF activation by examining the cell cycle-dependent chromatin association of each known MBF component. Once we establish that the Rep2 co-activator is constitutively bound to MBF throughout the cell cycle, while co-repressors dissociate during MBF activation, we elucidate how two key cell cycle regulators, CDK1 and the anaphase promoting complex/cyclosome (APC/C), modulate MBF activity to drive the transcription of genes required for DNA synthesis during S phase.

# Results

## Rep2 constitutively activates MBF-dependent transcription independently of cell cycle progression

To explore how different components of the MBF complex sense and regulate cell cycle progression, we deleted the activator rep2, the repressors yox1 or nrm1, and generated a double-deletion mutant strain (yox1Δrep2Δ). We then analyzed the expression of cdc22 and cdc18—two representative markers of MBF activity—in these strains. As shown in Fig. 1A, deletion of yox1 or nrm1 led to a dramatic fourfold to fivefold increase in the expression of cdc22 and cdc18. In contrast, deletion of rep2 significantly reduced the expression of these genes. Interestingly, in the double-deletion strain (yox1Δrep2Δ), which lacks both the activator and the repressor systems for MBF, the expression of cdc22 and cdc18 was largely similar to that of the wild-type strain.

To investigate how MBF activity is regulated in these genetic backgrounds, we synchronized cultures of wild-type, rep2Δ, yox1Δ, and rep2Δyox1Δ strains (Fig. EV1A) and analyzed the temporal expression of cdc22 and cdc18. As shown in Fig. 1B, MBF-dependent transcription in rep2Δ cells remained cell cycle-dependent, peaking at the G1/S transition, but at levels two- to four-fold lower than in the wild type. Conversely, yox1Δ cells exhibited constitutive activation of MBF-dependent transcription throughout the entire cell cycle, losing the characteristic peak at the G1/S transition. Intriguingly, the double mutant (rep2Δyox1Δ) displayed moderate, unregulated transcription of cdc22 and cdc18, lacking any clear cell cycle dependency. These findings suggested that Rep2 acts as a constitutive activator of MBF-dependent transcription, independent of cell cycle regulation. Thus, the repressor system comprising Nrm1 and Yox1 must not only repress MBF activity but also coordinate this repression with cell cycle progression. The critical importance of coordinating MBF activity with the cell cycle was further highlighted in co-culture experiments involving wild-type and yox1Δrep2Δ strains. In these experiments, the double mutant strain was outcompeted by wild-type cells (Fig. 1C), underscoring the fitness cost associated with dysregulated MBF activity.

To separate the different roles of Yox1 and Nrm1 in the repression of MBF-dependent transcription, we generated different chimeras, fusing each of them to Res2 at the res2 loci. As a control, we fused GFP to Res2 (Fig. EV1B). When we analyzed the expression of cdc18 or cdc22 in these strains, we noted that fusing GFP or Nrm1 (in the absence of the endogenous Nrm1) had minimal impact on the basal or induced (+HU) expression of cdc18 or cdc22, while having Rep2 fused to Res2 (in the absence of endogenous Rep2) minimally increased the basal expression of cdc22 (Fig. EV1C). On the contrary, cells expressing a Res2-Yox1 chimera showed a diminished capability to induce MBF-dependent genes after HU treatment, especially in the case of cdc22. In fact, Res2-Yox1 cells that had by-passed the function of Nrm1 since it was no longer required for loading Yox1 onto MBF, where unable to induce cdc18 or cdc22 during the G1/S transition either in cells with or without Nrm1 (Figs. 1D and EV1D), pointing that the primary repressor could be Yox1 and that Nrm1's role would be to facilitate the loading of Yox1 onto the MBF complex. Consistently with this lack of MBF regulation, cells with the Res2-Yox1 chimera were sensitive to drugs that induce replicative stress (Fig. EV1E) which correlates with a delayed septation peak in synchronous cultures when compared to wild type cultures, reflecting problems with the initiation of DNA replication (Fig. EV1D). To investigate potential functional overlap between Nrm1 and Yox1, we examined whether overexpression of either could produce a dominant phenotype. As shown in Fig. EV1F,G, Yox1 overexpression restored normal MBF-dependent transcription only in yox1Δ cells, but not in nrm1Δ or yox1Δnrm1Δ mutants. Similarly, Nrm1 overexpression rescued MBF-dependent transcription exclusively in nrm1Δ cells (Fig. EV1H,I). In summary, these results prompted us to consider the possibility that the binding and release of the repressor system Yox1/Nrm1 could be sufficient to regulate MBF activity in a cell cycle-dependent manner, while Rep2 would function merely as a co-activator of transcription involved solely in the amplitude of the MBF-dependent transcription and acting independently of cell cycle progression.

## Nrm1 is a sensor of cell cycle progression

The above findings led us to investigate whether Nrm1 or Yox1 could mediate the link between cell cycle progression and MBF-dependent transcriptional activity. To address this, we analyzed the binding dynamics of Cdc10 [a constitutive control previously shown to be persistently bound to MBF target promoters (Wuarin et al, 2002)], Rep2, Nrm1, and Yox1 to the cdc18 and cdc22 promoters during synchronized release from the G2/M transition. As shown in Fig. 2A and Appendix Fig. S1, while Cdc10 and Rep2 exhibited constitutive binding to the promoters, Nrm1 and Yox1 binding was tightly cell cycle-regulated. Their maximum promoter association peaked during G2, coinciding with minimal transcription of cdc18 and cdc22 (Fig. 1D), and was reduced from late mitosis to the end of S phase, when MBF activity was highest.

Parallel western blot analyses revealed that the protein levels of Yox1, Cdc10, and Rep2 remained relatively constant throughout the cell cycle (Fig. 2B). In contrast, Nrm1 levels declined sharply as cells entered mitosis in the two cell cycles that were analyzed, coinciding with the activation of MBF-dependent transcription.

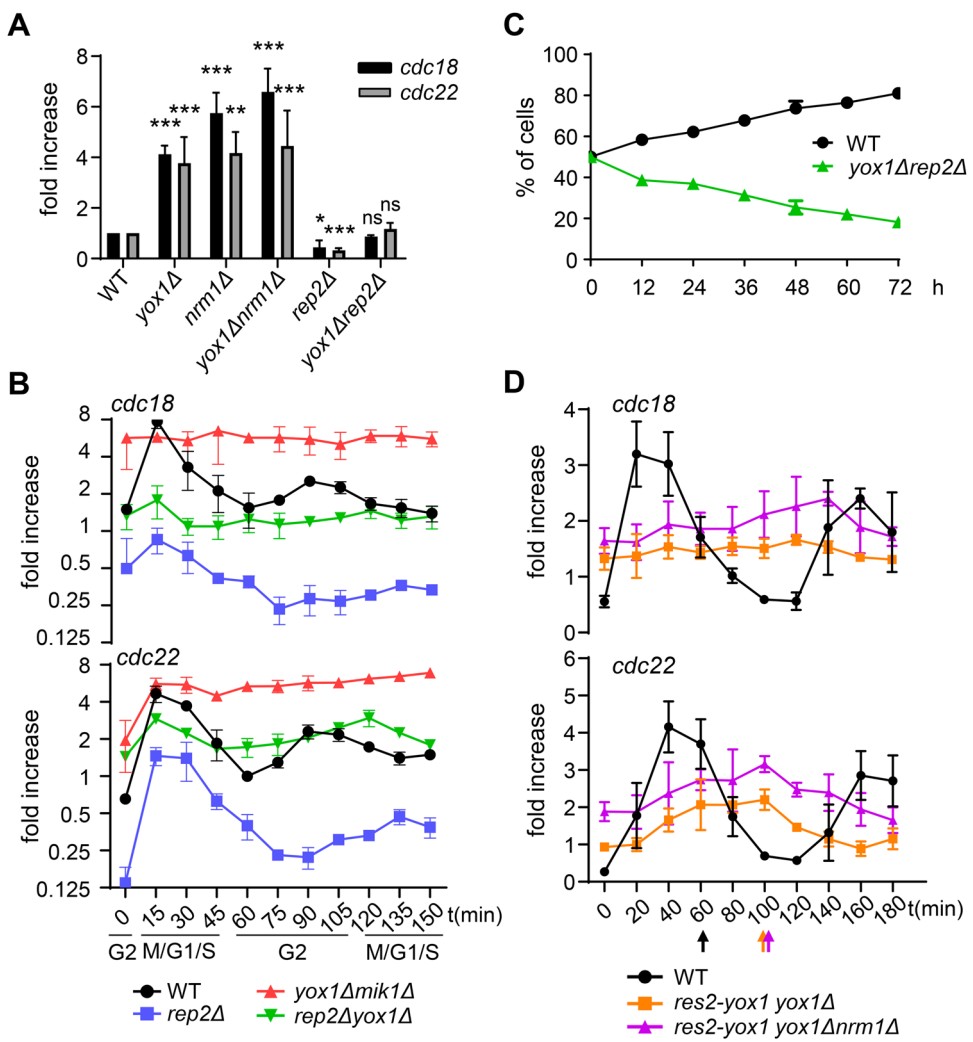

**Figure 1. MBF regulation is achieved solely through the repressor system Nrm1/Yox1.**

(A) qPCR of *cdc18* and *cdc22* expression in asynchronous cultures of WT, *yox1Δ*, *nrm1Δ*, *yox1Δnrm1Δ*, *rep2Δ* and *yox1Δrep2Δ* relative to WT expression level. *tfb2* was used as control gene. Graphic represents mean ± SD of *n* = 5 experiments. Statistics show significance from a Student's *T* test. *p < 0.05; **p < 0.01; ***p < 0.001. p(cdc18 expression against WT): 0.002334, 0.000725, 0.001313, 0.024752, and 0.270978, respectively. p(cdc22 expression against WT): 0.02144, 0.003493, 0.000537, 0.000562 and 0.658293, respectively. (B) qPCR of *cdc18* and *cdc22* expression in a *cdc2-asM17* block and release of WT (black), *yox1Δmik1Δ* (red), *rep2Δ* (blue) and *yox1Δrep2Δ* (green) relative to asynchronous WT expression level. *tfb2* was used as control gene. Graphic represents mean ± SD of *n* = 3 experiments. (C) Cultures of WT (expressing YFP) or *rep2Δyox1Δ* (expressing mRFP) cells were co-cultured and diluted during 72 h to keep cells exponentially growing. Cells were harvested every 12 h and analyzed by FACS. Graphic represents mean ± SD of *n* = 3 experiments. (D) qPCR of *cdc18* and *cdc22* expression in a *cdc25-22* block and release of WT (black), *res2-yox1 yox1Δ* (orange) and *res2-yox1 yox1Δ nrm1Δ* (purple). Res2-yox1 chimera is localized at the *res2* locus. Septation peak for each strain is depicted as an arrow below the *x* axis. *tfb2* was used as control gene. Graphic represents mean ± SD of *n* = 3 experiments relative to WT asynchronous expression. Source data are available online for this figure.

Interestingly, a higher-mobility Nrm1 band also appeared at this stage (marked with an asterisk in Fig. 2B), suggesting a post-translational modification. We decided to characterize the possible post-translational regulation of Nrm1 since it was already described to be phosphorylated under replicative stress (de Bruin et al, 2008; Dutta et al, 2008). To explore this, we examined whether Nrm1 was phosphorylated during a normal cell cycle. Extracts from metaphase- or hydroxyurea (HU)-arrested cells showed that Nrm1 exhibited a mobility shift indicative of phosphorylation during metaphase, but only a modest shift under HU-induced replicative stress (Fig. 2C). Treatment with phosphatase restored the mobility of Nrm1 (Fig. 2D), confirming the mobility shift was

due to phosphorylation. Furthermore, this phosphorylation-dependent shift was almost abolished in metaphase-arrested cells expressing a CDK1-analogue sensitive mutant (*cdc2-asM17*), only after the addition of 1-NM-PP1, implicating that CDK1 is directly or indirectly involved in Nrm1 phosphorylation (Fig. 2E).

Despite being a rather small protein containing only 342 residues, Nrm1 has multiple putative phosphorylation sites, including some predicted to be phosphorylated at specific phases of the cell cycle (Swaffer et al, 2018) and others potentially phosphorylated under replicative stress (de Bruin et al, 2008). Nrm1 contains 13 putative CDK1 phosphorylation sites (S/T-P) distributed across the protein (Fig. 3A). To identify the sites

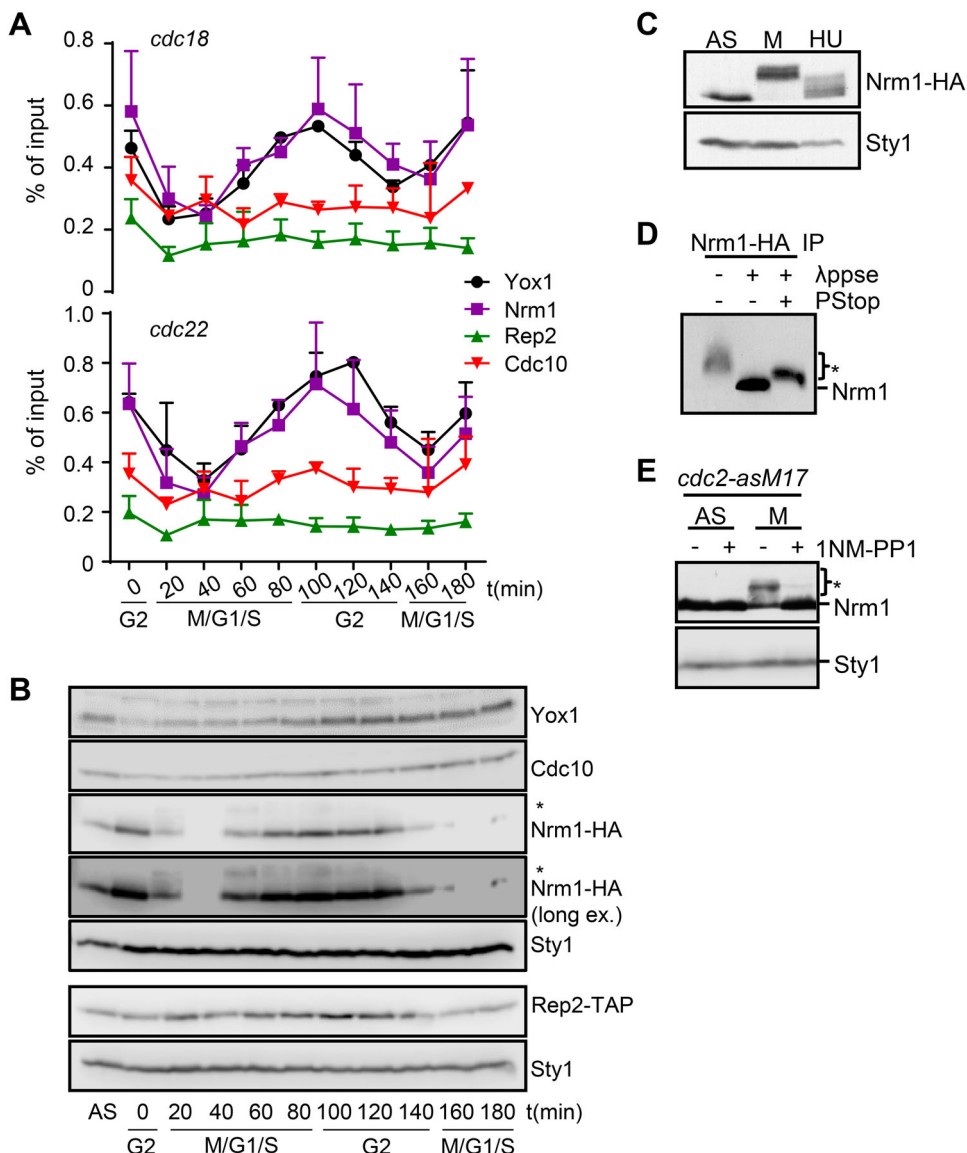

**Figure 2. Nrm1 is phosphorylated in metaphase.**

(A) ChIP analysis in *cdc18* and *cdc22* promoters of Yox1 (black), Nrm1-HA (purple), Rep2-TAP (green) and Cdc10 (red) from a *cdc25-22* block and release. Chromatin was immunoprecipitated with anti-HA (Nrm1), anti-Yox1, anti-PAP (Rep2-TAP) or anti Cdc10 antibodies. Plots represent the mean ± SD of at least $n = 3$ experiments. (B) Western blot of Yox1, Cdc10, Nrm1-HA and Rep2-TAP from a *cdc25-22* block and release. Sty1 is shown as loading control. Representative western blot of at least three different experiments are shown. (C) Western blot of extracts prepared from *nda3-KM311* strain expressing Nrm1-HA; asynchronous culture (AS), cells arrested in metaphase for 5 h at 18 °C (M), or cells treated with 10 mM Hydroxyurea (HU) for 3 h at 30 °C. Sty1 is shown as loading control. Representative western blot of at least three experiments are shown. (D) Native extracts (2 mg) from cells *nda3-KM311* arrested in metaphase were immunoprecipitated with anti-HA antibodies, treated with λ phosphatase were indicated (λppse) in the presence or absence of phosphatase inhibitors (PStop). The asterisk indicates the phosphorylated forms of Nrm1. Representative western blot of at least three experiments are shown. (E) Western blot of extracts prepared from *cdc2-asM17 nda3-KM311* strain expressing Nrm1-HA; asynchronous culture (AS), cells arrested in metaphase for 5 h at 18 °C (M). Where indicated, cells were treated with 1 μM 1NM-PP1 for 15 min to inhibit Cdc2. Sty1 is shown as loading control. The asterisk indicates the phosphorylated forms of Nrm1. Representative western blot of at least three experiments are shown. Source data are available online for this figure.

phosphorylated in metaphase, we purified Nrm1 from metaphase-arrested cells and performed mass spectrometry (MS). Phosphorylation was detected at 9 of the 13 CDK consensus sites, 5 of which had been previously reported during mitosis (Swaffer et al, 2018), validating our approach (Fig. EV2). To examine the functional impact of phosphorylation, we generated several Nrm1 mutants in which we were gradually replacing the putative phosphorylation sites to alanine (Nrm1-3A, Nrm1-4A, Nrm1-5A and Nrm1-6A). However, although we were able to decrease the mobility shift of Nrm1 in mitotically arrested cells, we could not completely abolish its phosphorylation (Appendix Fig. S2). Finally, we generated two Nrm1 mutants at eleven sites: a non-phosphorylatable mutant *nrm1-SA* (T9A T40A S57A T116A S155A S206A S237A T241A T272A T287A T307A) in which all 11 fully conserved CDK1

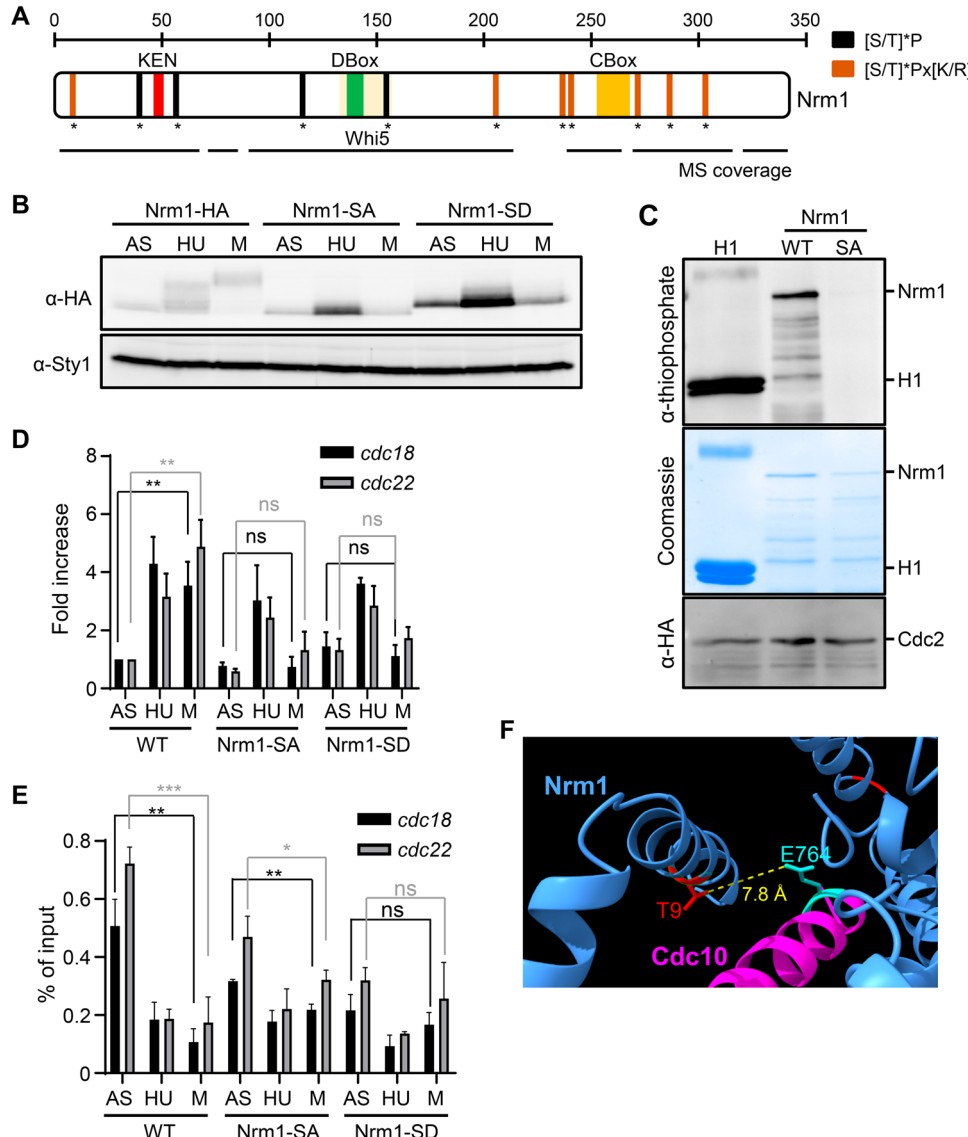

**Figure 3. Nrm1 is released from MBF when phosphorylated.**

(A) Schematic of Nrm1. The cream square shows the Whi5/Nrm1 repressing domain. Consensus APC/C motifs are shown: KEN box (48–50, red), overlapping DBox (138–141 and 140–143, green) and CBox (251–255, 266–270, yellow). The putative phospho-serine/threonine are shown (in orange, those with the strict consensus, S/T-P-X-R; in black: S/T-P). The solid line shows the coverage in the M/S experiment and the asterisks indicate the residues that were mutated in the Nrm1-SA/SD mutants. (B) Western blot analysis of Nrm1-HA Nrm1-SA-HA and Nrm1-SD-HA in the *nda3-KM311* background. Cells were grown and treated as in Fig. 2C. AS asynchronous culture; HU, cells treated with 10 mM HU for 3 h at 30 °C; M, cells arrested in metaphase for 5 h at 18 °C. Sty1 is shown as loading control. A representative western blot of at least three experiments are shown. (C) In vitro kinase assay on human histone H1 (positive control), Nrm1 WT and Nrm1-SA purified from *E. coli*. Cdc2-HA was immunoprecipitated with α-HA beads from 250 μg of whole cell extracts (WCE). Half of the kinase reaction was analyzed by western blotting (α-thiophosphate and α-HA) and the other half was used for Coomassie staining. Representative gels of at least three different experiments are shown. (D) qPCR transcription analysis of *cdc18* and *cdc22* expression in the Nrm1-HA, Nrm1-SA-HA and Nrm1-SD-HA in the *nda3-KM311*. Cells were grown and treated as in Fig. 2B. AS asynchronous culture; HU cells treated with 10 mM Hydroxyurea for 3 h at 30 °C M; cells arrested in metaphase for 5 h at 18 °C. *Tfb2* was used as housekeeping gene. Plot represents mean ± SD of at least $n = 3$ experiments. Statistics show significance from a Student's $T$ test. ns: $p > 0.05$; *$p < 0.05$; **$p < 0.01$; ***$p < 0.001$. $p$(*cdc18* expression AS vs M in WT, Nrm1-SA and Nrm1-SD): 0.003048, 0.737168 and 0.594176, respectively. $p$(*cdc22* expression AS vs M in WT, Nrm1-SA and Nrm1-SD): 0.03399, 0.09294 and 0.268925, respectively. (E) ChIP analysis of Nrm1-HA, Nrm1-SA-HA and Nrm1-SD-HA in nda3-KM311 strain. ChIP was done as described with anti-HA antibodies. The isolated DNA was used to amplify the promoter region of *cdc18* and *cdc22*. Cells were grown and treated as in Fig. 2C. AS asynchronous culture; HU, cells treated with 10 mM Hydroxyurea for 3 h at 30 °C M; cells arrested in metaphase for 5 h at 18 °C. Plot represents mean ± SD of at least $n = 3$ experiments. Statistics show significance from a Student's $T$ test. ns: $p > 0.05$; *$p < 0.05$; **$p < 0.01$; ***$p < 0.001$. $p$(Nrm1 ChIP to *cdc18* promoter AS vs M in WT, Nrm1-SA and Nrm1-SD): 0.002549, 0.001104 and 0.366253, respectively. $p$(Nrm1 ChIP to *cdc22* promoter AS vs M in WT, Nrm1-SA and Nrm1-SD): 0.000816, 0.031564 and 0.456596, respectively. (F) Alphafold prediction of the interaction between Nrm1 (blue) and Cdc10 (magenta) in the complete MBF complex. Nrm1 CDK consensus sites are depicted in red. Glutamic acid 764 from Cdc10 is indicated in cyan. Source data are available online for this figure.

consensus sites (9 identified by MS plus 2 additional sites without MS peptide coverage) were replaced to alanine; and a phosphomimetic mutant (*nrm1-SD*), in which these sites were replaced by aspartic or glutamic acid. Consistent with our MS data, the metaphase-associated mobility shift was abolished in *nrm1-SA* cells (Fig. 3B). *nrm1-SD* cells, in which it was also abolished the metaphase mobility shift, had a different basal mobility compared to the wild type cells (Fig. 3B, last 3 lanes), probably due to the increased negative charge of the Nrm1-SD when compared to unphosphorylated wild type Nrm1. We also confirmed that Cdc2 was able to in vitro phosphorylate bacterially produced Nrm1, but was unable to phosphorylate the mutant Nrm1-SA (Fig. 3C), supporting our observation that Nrm1-SA did not have any change in electrophoretic mobility in mitotically arrested cells (Fig. 3B).

To assess the impact of these mutants on MBF-dependent transcription, we arrested cells in metaphase (M) or S phase (HU treatment) and monitored *cdc18* and *cdc22* expression. In *nrm1-SA* cells, MBF-dependent transcription failed to activate in metaphase but was induced under HU-arrest, likely due to phosphorylation of Yox1 by the replication checkpoint kinase Cds1 (Gómez-Escoda et al, 2011). Similarly, *nrm1-SD* cells also failed to activate MBF-dependent transcription in metaphase, although basal transcription levels were modestly elevated compared to *nrm1-SA* and wild-type cells (Fig. 3D and Appendix Fig. S3). These results highlight the critical role of CDK1-mediated Nrm1 phosphorylation in metaphase-specific MBF activation, independently of the replication checkpoint pathway.

To investigate why *nrm1-SA* and *nrm1-SD* cells failed to activate MBF-dependent transcription in metaphase, we performed chromatin immunoprecipitation (ChIP) to analyze the binding of Nrm1 and its mutants at *cdc18* and *cdc22* promoters under various conditions. In wild-type cells, Nrm1 dissociated from these promoters in metaphase or HU-arrested cells, correlating with transcriptional activation. However, in *nrm1-SA* cells, Nrm1-SA remained bound to these promoters in metaphase, while *nrm1-SD* cells exhibited reduced basal binding to these promoters, but failed to fully dissociate completely in metaphase-arrested cells (Fig. 3E and Appendix Fig. S4). These results confirm that CDK1-dependent phosphorylation is essential for Nrm1 release from chromatin, enabling MBF activation during metaphase. Interestingly, structural modeling of the complete MBF complex using AlphaFold (Abramson et al, 2024), revealed that threonine 9 in Nrm1 is positioned near glutamate 764 of Cdc10 (Fig. 3F). This proximity could explain why Nrm1 loses its interaction with the MBF complex upon phosphorylation, although experiments with mutations in *trans* would be required to determine if this is this hypothesis is true.

To obtain a wider picture of the effect of these Nrm1 mutants on MBF-dependent transcription, we synchronized cells (Fig. EV3A) and collected protein and RNA samples every 20 min over two cell cycles. In wild-type cells (Nrm1 WT), a higher-mobility Nrm1 band was noticeable after mitosis (marked with an asterisk in Fig. 4A). In contrast, neither *nrm1-SA* nor *nrm1-SD* cells exhibited cell cycle-dependent fluctuations or mobility shifts (Fig. 4A). RNA sequencing (RNAseq) of biological duplicates of each time-point revealed minimal effects on the overall transcriptome. However, cell cycle regulation of 43 genes with expression profiles similar to *cdc18* or *cdc22* was abolished in both *nrm1-SA* and *nrm1-SD* mutant cells (Fig. EV3B,C). Notably, 21 of these 43 genes, 21 were previously

shown to have Cdc10, Res1 and Res2 ChIP-seq peaks in their promoter region (Aligianni et al, 2009b; Skribbe et al, 2025b). This subset of 21 genes showed a pronounced loss of cell cycle regulation (Fig. 4B), with the effect being more pronounced in *nrm1-SA* cells when compared to *nrm1-SD* (Figs. 4C and EV3C).

In conclusion, our findings establish that Nrm1 is a critical target of CDK1-mediated cell cycle regulation, orchestrating MBF-dependent transcription through phosphorylation-dependent release from chromatin. By coupling transcriptional control to cell cycle progression, Nrm1 phosphorylation ensures precise timing of MBF-dependent gene expression.

## Nrm1 is degraded by APC/C

While analyzing the protein extracts, we consistently observed higher levels of Nrm1-SD compared to wild-type Nrm1 or Nrm1-SA (Fig. 5A). To explore this further, we measured the half-life of the three proteins. While the half-life of Nrm1-SA (19 min) was comparable to that of wild-type Nrm1 (13 min), the half-life of Nrm1-SD was significantly longer (119 min) (Fig. 5B).

Given that Nrm1 in budding yeast is actively degraded by the proteasome (Ostapenko et al, 2012; Ostapenko and Solomon, 2011), we investigated whether the proteasome also degrades Nrm1 in fission yeast. Consistent with this hypothesis, Nrm1 was stabilized in the temperature-sensitive proteasome mutant *mts3-1* (Fig. 5C). Moreover, ubiquitinated forms of Nrm1 accumulated when the proteasome was inactivated (Fig. 5D).

To identify degradation signals within Nrm1, we analyzed its sequence for conserved degradation motifs. We detected a putative KEN box (residues 48–50) and two overlapping DBox motifs (residues 138–141 and 140–143) (Fig. 3A). However, mutating these sites—either individually or simultaneously—did not fully prevent Nrm1 degradation via the proteasome (Fig. 5E). We then divided Nrm1 into two halves: the N-terminal region (residues 1–173), which includes the KEN box and DBox motifs (Nrm1 A), and the C-terminal region (Nrm1 C), containing the last 172 residues (residues 171-342). Both fragments were degraded with similar half-lives (Fig. EV4A). Interestingly, mutating the KEN box and DBox motifs in the N-terminal fragment stabilized the protein, suggesting that while these motifs play a role in targeting Nrm1 for degradation, additional elements in the C-terminal region also contribute (Fig. 5E and Appendix Fig. S5).

The alignment of the Nrm1 sequence from *S. pombe* with other *Schizosaccharomyces* species revealed two conserved five-amino-acid motifs in the C-terminal domain (residues 251–255 and 266–270). To evaluate the contribution of these domains to Nrm1 stability, we introduced mutations in these motifs—either replacing all ten residues with alanines or introducing point mutations in two residues of each motif—and measured the half-life of the resulting mutants. These experiments were performed in the context of both the isolated C-terminal domain (Fig. EV4B) and the full-length Nrm1 combined with KEN box and DBox mutations (Fig. 5F and Appendix Fig. S5). In both cases, the mutations significantly stabilized Nrm1, confirming the existence of three independent degradation signals—two in the N-terminal region (KEN and DBox) and one in the C-terminal region (that we termed CBox)—that collectively regulate Nrm1 stability.

Finally, to determine whether Nrm1 degradation was indeed mediated by the Anaphase-Promoting Complex/Cyclosome

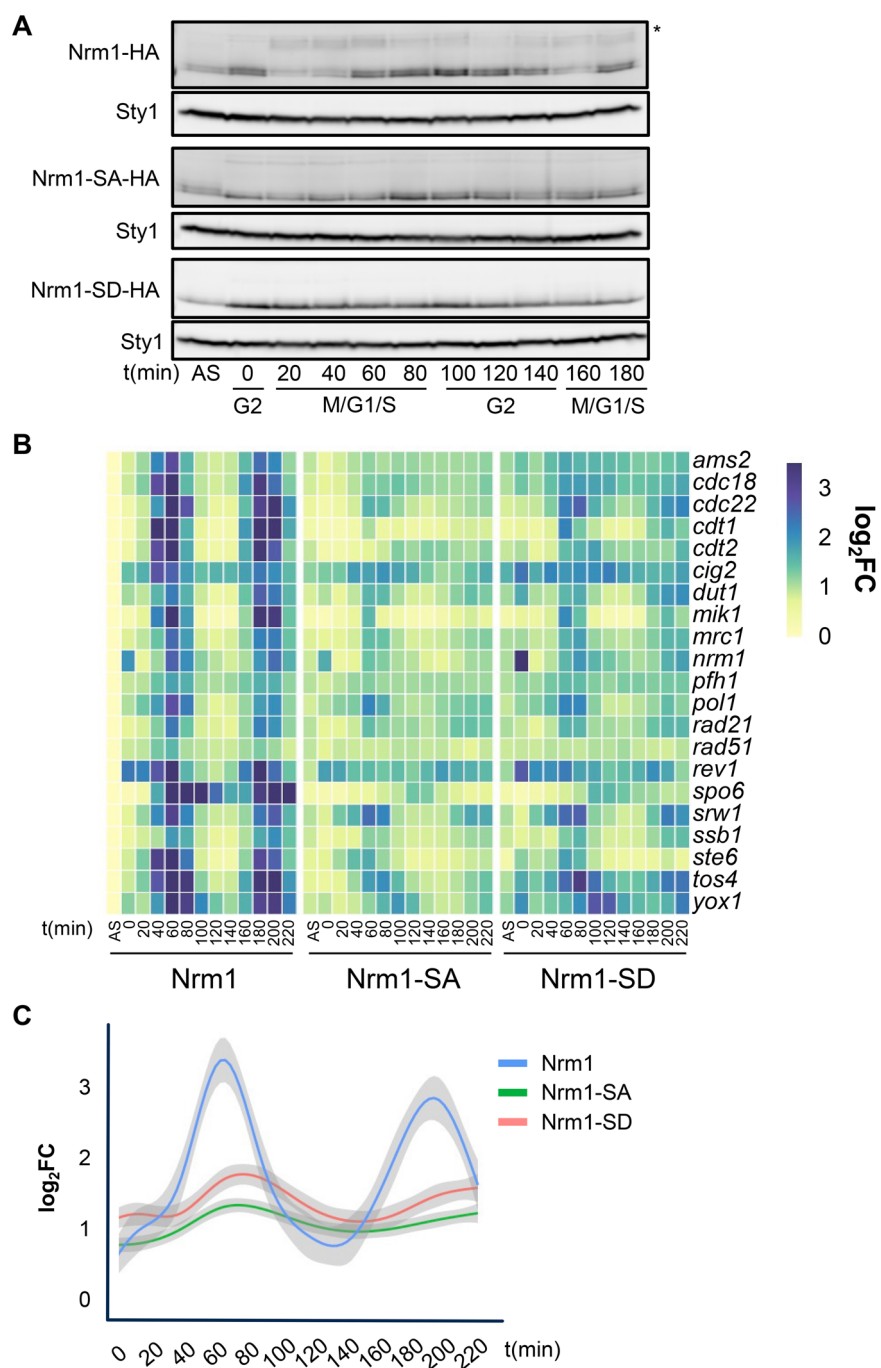

**Figure 4. Nrm1 phosphorylation is essential for correct MBF-dependent transcription.**

(A) Western blot analysis of Nrm1-HA, Nrm1-SA-HA and Nrm1-SD-HA in a *cdc25-22* block and release. The asterisk indicates Nrm1 phosphorylation forms. Sty1 is shown as a loading control. AS: asynchronous. Representative Western blots of at least three different experiments are shown. (B) Heatmap of gene expression of 21 MBF-dependent genes. RNAseq of a wild type (Nrm1), Nrm1-SA and Nrm1-SD in a *cdc25-22* block and release. Genes were selected as cycling genes peaking transcription at 60–80 and 180–200 min after release and having expression altered in Nrm1-SA and Nrm1 SD (Fig. EV3B). This list was crossed with the MBF ChIP data published before (Aligianni et al, 2009b; Data ref: Aligianni et al, 2009a; Skribbe et al, 2025b, Data ref: Skribbe et al, 2025a) to further delimit candidates. Scale represents Log$_2$FC of normalized reads of each time-point relative to Nrm1 asynchronous (AS) values. (C) Expression of MBF-dependent genes from (C) in a Nrm1 (blue), Nrm1-SA (green) and Nrm1-SD (red). Line represents mean of values and grey shadow represents SD. Source data are available online for this figure.

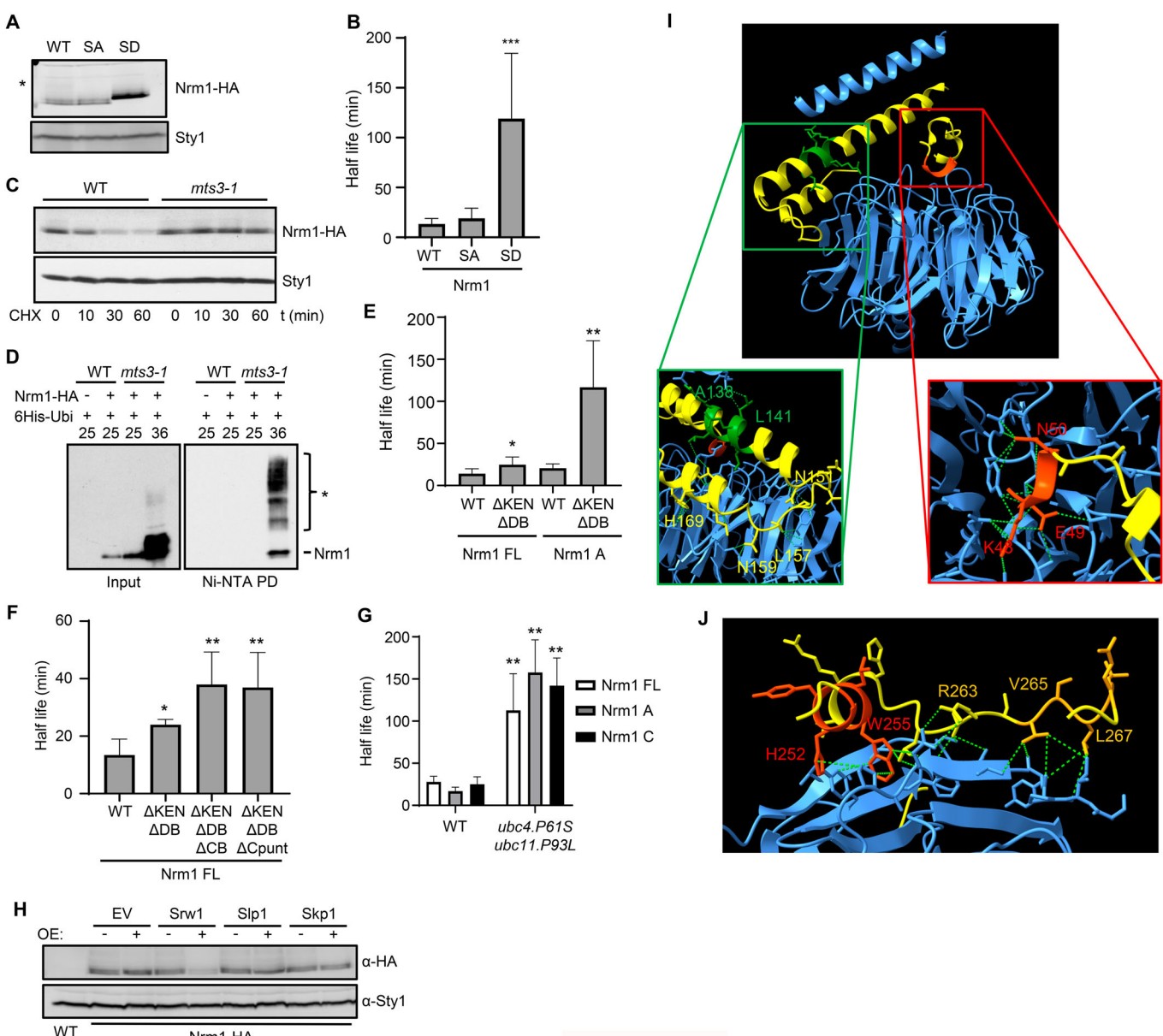

(APC/C), we measured its half-life in the two E2 ubiquitin-conjugating enzyme mutants of the APC/C in fission yeast, *ubc4.P61S* and *ubc11.P93L*. As shown in Fig. 5G, Nrm1 was stabilized in these mutants, with Ubc4 identified as the primary E2 enzyme driving Nrm1 degradation (Fig. EV4C). To identify which APC/C activator mediates Nrm1 degradation, we overexpressed the two mitotic APC/C activators (Slp1 or Srw1/Ste9) and analyzed Nrm1 stability by western blot. We overexpressed the SCF activator Skp1 as a negative control. As shown in Fig. 5H, only over-expression of Srw1 resulted in fast degradation of Nrm1, suggesting that APC/C$^{Srw1}$ might be responsible for the degradation of Nrm1. Structural modeling of Srw1 and Nrm1 using AlphaFold revealed potential interactions between Srw1 and the 3 proposed degradation motifs on Nrm1, the KEN box, the DBox (Fig. 5I), and the newly identified recognition site, the CBox (Fig. 5J), which would support our results suggesting that Nrm1 contains 3 different elements that need to be coordinately recognized by APC/C.

Interestingly, we have shown that Srw1 is an MBF-dependent gene (Fig. 4B). In fact, we were able to detect 2 MCB elements in the promoter of *srw1* and we were able to ChIP Cdc10 on both sites, confirming that *srw1* is an MBF-regulated gene (Fig. EV4D). This discovery highlights a novel and redundant mechanism of negative feedback regulation of MBF activity. In this mechanism, MBF indirectly regulates its own activity by controlling the expression of *srw1*, which then promotes APC$^{Srw1}$-dependent degradation of Nrm1. Remarkably, while Nrm1-SA is an excellent substrate of Srw1, Nrm1-SD is highly stable even when Srw1 was overexpressed (Fig. EV4E).

## Cellular fitness requires dynamic phosphorylation and dephosphorylation of Nrm1

Our findings demonstrate that both the phosphorylation-deficient *nrm1-SA* mutant and the phosphomimetic *nrm1-SD* mutant exhibit

◀ **Figure 5. Nrm1 is degraded by the ubiquitin–proteasome pathway.**

(A) Western blot analysis of asynchronous (AS) Nrm1-HA, Nrm1-SA-HA and Nrm1-SD-HA protein extracts from Fig. 4A. Sty1 is shown as loading control. Representative western blots of at least three experiments are shown. (B) Half-life in minutes of Nrm1 (pAY1100), Nrm1-SA (pAY1149) and Nrm1-SD (pAY1150) after thiamine and cycloheximide (CHX) addition. TCA samples were collected at 0, 10, 30 and 60 min after cycloheximide addition and assayed for western blot. Sample loading was normalized with Sty1. Half-life was calculated using GraphPad Prism. Plot represents mean ± SD of at least $n = 3$ experiments. Statistics show significance from a Student's T test. *$p < 0.05$; **$p < 0.01$; ***$p < 0.001$. p(Nrm1-HA vs Nrm1-SA-HA or Nrm1-SD-HA): 0.273774 and 0.001689, respectively. (C) Western blot of extracts prepared from wild type (WT) or mts3-1 cells. At the bottom is indicated the time after addition cycloheximide (CHX) and when cells were changed to non-permissive temperature (36 °C). Sty1 is shown as loading control. Representative western blots of at least three experiments are shown. (D) Alkaline denaturing extracts were prepared from wild type (WT) or mts3-1 cells and subjected to a pull down (PD) with Ni-NTA beads. Cells, with or without Nrm1-HA, were expressing 6xHis-ubiquitin from the nmt1 promoter (His-Ubi) and, when indicated, were shifted to 36 °C for 4 h. Input (80 µg) and pull downs were analyzed on a 10% polyacrylamide gel. Western blot detection was performed with anti-HA antibodies. The asterisk indicates the ubiquitinated forms of Nrm1. Representative western blots of at least three experiments are shown. (E) Half-life in minutes of full length (Nrm1 FL) Nrm1 (WT), Nrm1 ΔKEN ΔDBox (ΔKENΔDB), the first half of Nrm1 containing the KEN and DBox (Nrm1 A, WT) or with mutated KEN and DBox (ΔKENΔDB). Extracts were prepared after thiamine and cycloheximide addition. TCA samples were collected at 0, 10, 30 and 60 min after cycloheximide addition and essayed for western blot. Sample loading was normalized with Sty1. Half-life was calculated using GraphPad Prism. Plot represents mean ± SD of at least $n = 3$ experiments. Statistics show significance from a Student's T test. *$p < 0.05$; **$p < 0.01$; ***$p < 0.001$. p(Nrm1 FL WT vs Nrm1 FL ΔKEN ΔDBox): 0.045366. p(Nrm1 A WT vs Nrm1 A ΔKEN ΔDBox): 0.002664. (F) Half-life in minutes of full length Nrm1, Nrm1 ΔKEN ΔDBox (ΔKENΔDB), Nrm1 ΔKEN ΔDBox ΔCBox (ΔKEN ΔCB) and Nrm1 ΔKEN ΔDBox ΔCpunt (H252A W255A, R263A, V265A, L267A). Extracts were prepared after thiamine and cycloheximide addition at 0, 10, 30 and 60 min and analyzed in western blot. Sample loading was normalized with Sty1. Half-life was calculated using GraphPad Prism. Plot represents mean ± SD of at least $n = 3$ experiments. Statistics show significance from a Student's T test. *$p < 0.05$; **$p < 0.01$; ***$p < 0.001$. p(Nrm1 FL WT vs ΔKEN ΔDBox, ΔKEN ΔDBox ΔCB or ΔKEN ΔDBox ΔCpunt): 0.014104, 0.001357 and 0.002309, respectively. (G) Half-life in minutes in WT and ubc4.P61S ubc11.P93L cells of Nrm1 full length (WT), the first 173 residues of Nrm1 containing the KEN and DBox (Nrm1 A) and the carboxi-region of Nrm1 containing the last 171 residues (Nrm1 C). Cultures were grown at 25 °C and incubated 1 h at 36 °C before cycloheximide addition. TCA samples were collected at 0, 10, 30 and 60 min after the temperature shift and analyzed in western blot. Sample loading was normalized with Sty1. Half-life was calculated using GraphPad Prism. Plot represents mean ± SD of at least $n = 3$ experiments. Statistics show significance from a Student's T test. *$p < 0.05$; **$p < 0.01$; ***$p < 0.001$. p(WT vs ubc4.P61S ubc11.P93L in Nrm1 FL, Nrm1 A and Nrm1 C): 0.010782, 0.003324 and 0.00399, respectively. (H) Western blot analysis of extracts from Nrm1-HA strains with an empty plasmid (EV), or with plasmids overexpressing Srw1, Slp1 or Skp1. Cells were grown in EMM supplemented with 20 µM thiamine (OE −), or grown in EMM (OE +) to allow expression. Sty1 is shown as loading control. Representative western blots of at least three experiments are shown. (I) Alphafold prediction of the interaction between Nrm1 (yellow) and Srw1 (blue). The inset shows predicted contacts between Nrm1 KEN box (red) or Nrm1 DBox (green) with Srw1. (J) Alphafold prediction of the interaction between Nrm1 CBox (yellow) with Srw1 (blue). Predicted Nrm1 contact residues are colored in red (251–255) or orange (265–270). Source data are available online for this figure.

similar transcriptional outcomes (Fig. 4BC). In both cases, the connection between cell cycle progression and MBF-dependent transcription is disrupted. The key distinction lies in basal transcription levels: while MBF-dependent transcription in the nrm1-SA strain remains tightly repressed, the nrm1-SD strain shows a moderately elevated basal transcription levels and retains a residual, albeit minimal, degree of cell cycle-dependent transcriptional regulation (Fig. 4C). Next, and to determine whether the nrm1-SA or nrm1-SD mutations affected the cellular response to DNA damage or increased replication stress, we further characterized both strains. As shown in Fig. EV5A, both mutants responded similarly to hydroxyurea (HU) treatment, and neither displayed Yox1 phosphorylation under normal conditions -a marker indicative of replicative stress. Consistent with this, both nrm1-SA and nrm1-SD strains exhibited wild-type growth on solid media containing increasing concentrations of HU or methyl methanesulfonate (MMS) (Fig. EV5B). However, in liquid culture, we observed a subtle but reproducible growth defect, more pronounced in the nrm1-SA strain (Fig. EV5C). To directly assess DNA damage, we quantified Rad52 foci in asynchronous cultures. The nrm1-SA strain showed an increased proportion of cells with Rad52 foci, suggesting elevated DNA damage levels. This phenotype closely mirrors that of the rad3Δ strain (Fig. EV5D).

To assess how these alterations impact cellular fitness, we performed competitive co-culture experiments involving wild-type, nrm1-SA, and nrm1-SD strains. As shown in Fig. 6A–C, both mutant strains were equally outcompeted by the wild-type strain, highlighting the essential role of dynamically regulated MBF-dependent transcription in cellular fitness. Moreover, when nrm1-SA and nrm1-SD strains were co-cultured with each other, neither mutant exhibited a fitness advantage, further emphasizing that

fixing Nrm1 phosphorylation status -whether constitutively phosphorylated or unphosphorylated- results in equivalent fitness defects. To investigate the decreased cellular fitness observed in the nrm1-SA and nrm1-SD strains and determine whether these mutations impact any specific phase of the cell cycle, we introduced both mutations into a cell cycle reporter system that precisely measures the duration of each phase in fission yeast (Murciano-Julia et al, 2025). As shown in Fig. 6D, the overall length of the cell cycle was only slightly affected (150.63 ± 20.19 min in WT, 162.84 ± 32.68 min in Nrm1-SA, and 163.92 ± 19.31 min in Nrm1-SD). However, the combined duration of the M/G1 and S phases was significantly extended, with increases of 25% and 18%, respectively (48.46 ± 8.18 min in WT, 60.67 ± 12.97 min in Nrm1-SA, and 57.31 ± 10.97 min in Nrm1-SD). Notably, the delay in S-phase initiation was more pronounced in the Nrm1-SA strain (Fig. 6E).

These results underscore the critical importance of a dynamic and cell cycle-regulated interplay between Nrm1 phosphorylation and dephosphorylation for maintaining proper MBF-dependent transcription and ensuring cellular fitness. The disruption of this regulatory mechanism, irrespective of the specific direction, compromises the cell ability to synchronize transcription with cell cycle progression, leading to competitive disadvantages.

## Discussion

From yeasts (both budding and fission) to metazoans, the G1/S transcriptional wave is a highly conserved feature of the cell cycle, with substantial overlap in the genes it regulates. Additionally, the mechanisms that respond to DNA damage or challenges to DNA

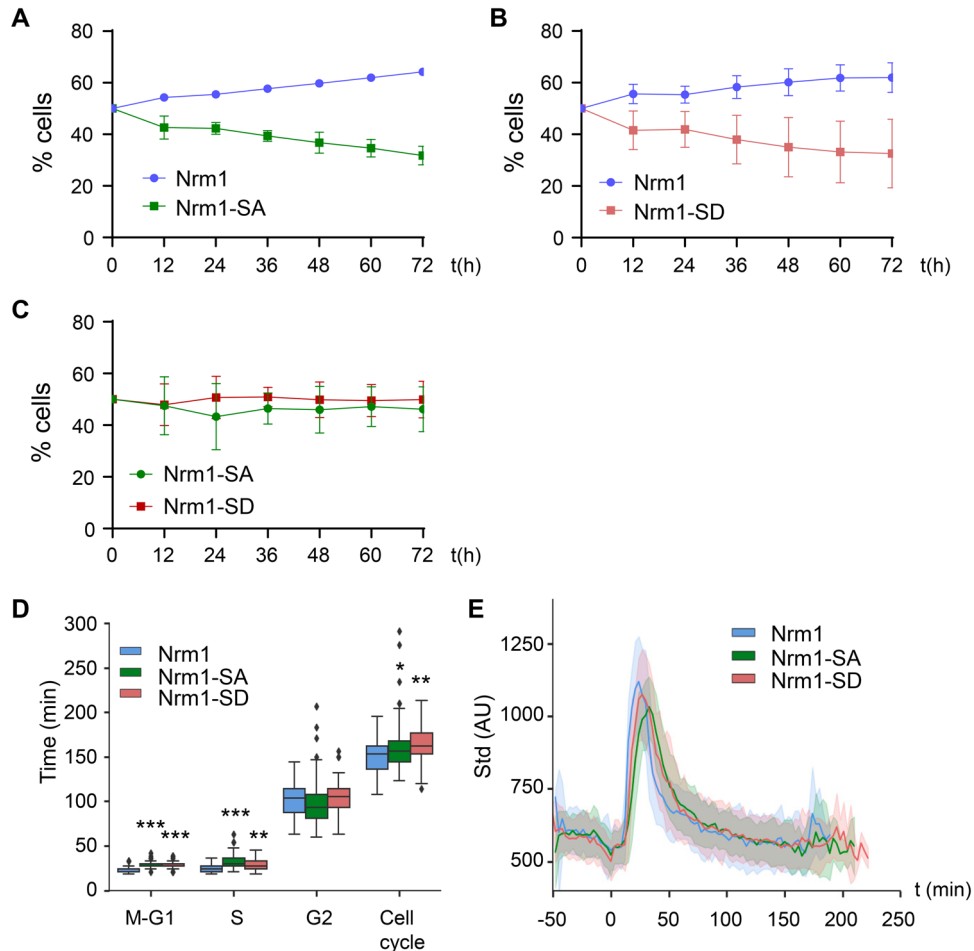

**Figure 6. Nrm1 phosphorylation is essential for cellular fitness.**

(A–C) Cultures of WT (Nrm1) vs Nrm1-SA (A), WT (Nrm1) vs Nrm1-SD (B) or Nrm1-SA vs Nrm1-SD (C) cells were co-cultured and diluted during 72 h to keep cells exponentially growing. Cells were harvested every 12 h and analyzed by FACS. All strains carried either eGFP-Pcn1 or mCherry-Pcn1. Experiments were conducted with the two-color combinations per competition. Two biological replicas were done per competition and in both color combinations. Final results for each competition were merged. Graphic represents mean ± SD of n = 4 biological replicas (2 eGFP vs mCherry tag and 2 mCherry vs eGFP tag). (D) Cell cycle quantification of wild type (Nrm1), Nrm1-SA and Nrm1-SD using the FLCCR reporter strain. Boxplots represent median and 25–75 quartiles and whiskers extend to the rest of the distribution. Outliers are depicted as dots. Statistics show significance from a Student's T test. *p < 0.05; **p < 0.01; ***p < 0.001. Nrm1 n = 38 cells, Nrm1-SA n = 53 cells, Nrm1-SD n = 56 cells. p(Nrm1 vs Nrm1-SA): 4.88E-09, 3.74E-05, 0.7893, 0.0443, respectively. p(Nrm1 vs Nrm1-SD): 4.74E-08, 0.0020, 0.3126, 0.0018 respectively. (E) Representation of mCherry-Pcn1 Std of wild type (Nrm1), Nrm1-SA and Nrm1-SD videos. Cell cycles are aligned to time = 0 (peak of SynCut3-mTagBFP2). Lines represent the mean of values and colored shades show SD of values. Source data are available online for this figure.

synthesis and their impact on the transcription factors driving the G1/S transcriptional wave are also well conserved (Bertoli et al, 2013a; Bertoli et al, 2013b; de Bruin et al, 2008; Gómez-Escoda et al, 2011). However, significant differences exist in how different species achieve G1/S transcriptional activation. In mammalian cells and budding yeast, activation occurs through phosphorylation of the repressor (RB or Whi5, respectively), which promotes its release from the transcription factor (E2F or SBF, respectively) [reviewed in (Bertoli et al, 2013b)], although an intermediate Rb-E2F activity state has been proposed in mammalian cells, which would be dependent on phosphorylations of Rb at specific residues (Konagaya et al, 2024). In budding yeast, where the G1/S transcriptional wave is under the control of two different factors, MBF and SBF, Whi5—the SBF repressor—is phosphorylated in a CDK-dependent manner and translocated to the cytoplasm

inducing the SBF-dependent transcription after START (Costanzo et al, 2004; de Bruin et al, 2004). Budding yeast MBF activity is constrained to G1 through feedback with its specific repressor Nrm1 (de Bruin et al, 2006). In contrast, fission yeast employs a more intricate mechanism. Despite possessing a Whi5 orthologue, this protein does not appear to play a major role in mitotic G1/S transcription regulation under normal growth conditions (Celia Gálvez-Merchán, Livia Pérez-Hidalgo and Sergio Moreno, personal communication). Instead, Nrm1 in fission yeast assumes a dual role: initiating the G1/S transcriptional wave in metaphase and switching it off at the end of S phase by reloading (together with Yox1) onto the MBF complex (Fig. 7A). Thus, fission yeast Nrm1 combines the regulation of budding yeast Whi5 and Nrm1 to constrain MBF activity from metaphase till the end of S phase. Importantly, Nrm1 alone has minimal repressive activity; it

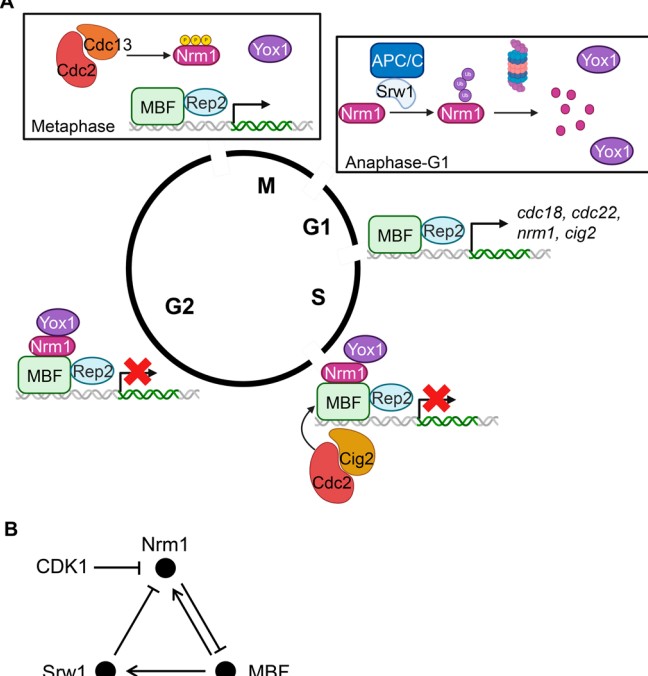

**Figure 7. Cell cycle-dependent MBF regulation.**

(A) Cartoon depicting the regulation of MBF during cell cycle. During G2 phase, MBF remains inactive due to the binding of Nrm1/Yox1. At metaphase, Cdc2-Cdc13 complex phosphorylates Nrm1 causing its release from MBF leading to pre-activated MBF. Unphosphorylated Nrm1 is then ubiquitinated by APC/C$^{Srw1}$ and degraded by the proteasome. Nrm1 then accumulates until it is able so repress again MBF at the end of S phase along with Res1 phosphorylation by Cdc2-Cig2. (B) Diagram showing the negative feedback loops regulating MBF activity. Srw1 plays an essential role to induce cell cycle-regulated transcription through the degradation of Nrm1. In the absence of Srw1 or in the absence of Nrm1 phosphorylation, the system becomes a simple feedback loop, with constitutive transcription.

primarily functions as a cell cycle–dependent sensor that facilitates the recruitment of Yox1—the true repressor—to the MBF complex. This mechanism represents a significant difference from the regulatory systems described in budding yeast and higher eukaryotes.

Our findings reveal that S-phase transcription in fission yeast is induced by alleviating repression from the Nrm1/Yox1 system, while the activator Rep2 remains constitutively bound to the MBF complex. This activation unfolds through a two-step repression-relief mechanism. The first step involves CDK-dependent phosphorylation of Nrm1 during metaphase, which temporarily releases Nrm1 from the MBF complex. This transient and reversible activation serves as a "kick-start" to MBF-dependent transcription, ensuring the accumulation of critical factors like Srw1/Ste9, which is an MBF-dependent gene itself. The second step occurs later, enabling APC/C$^{Srw1/Ste9}$ to degrade unphosphorylated Nrm1. This irreversible step amplifies and sustains MBF-dependent transcription, maintaining it until the repressor system (Nrm1/Yox1 and Cig2) is reassembled in late S phase when APC/C activity is inactivated. Importantly, this second step has an additional layer of regulation. Both Nrm1 and Srw1 are themselves MBF-regulated genes. As Srw1 levels increase due to MBF activation, more Nrm1 is

targeted for degradation by APC/C$^{Srw1}$, creating a positive feedback loop. This feedback prevents MBF activity from shutting down prematurely and sustains transcription until the end of S phase (Fig. 7B). At this point, APC/C is inactivated, allowing Nrm1 to stabilize, rebind to MBF, and repress MBF-dependent transcription from the end of S phase. It is important to note that, without the intervention of APC/C$^{Srw1}$, a tightly regulated system like this would function as a perfect negative feedback loop, akin to a centrifugal governor. While this setup would produce strictly regulated, constitutive MBF activity, it would not be able to change its output dynamically along the cell cycle.

Intriguingly, mammalian cells appear to have evolved a comparable mechanism (Yao et al, 2008). The APC/C has been shown to degrade the atypical repressive E2Fs (E2F7 and E2F8) (Boekhout et al, 2016). It would be compelling to investigate whether activating E2Fs in mammals also promote the transcription of *CDH1*, analogous to the role of Srw1 in fission yeast, as our results suggest that this is essential for establishing the G1/S transcriptional wave. In the absence of Nrm1 degradation—whether due to APC/C inactivation or because Nrm1 remains chromatin-bound (as in the Nrm1-SA mutant)—MBF-dependent transcription becomes tightly regulated through negative feedback but loses its cell cycle dependency. This loss of regulation results in fitness defects, as such cells are readily outcompeted by wild-type counterparts.

Additionally, our data suggest that Nrm1 is not a preferred substrate for Srw1-dependent degradation. While Nrm1 turnover requires Srw1, and Srw1 overexpression significantly shortens Nrm1 half-life, partial Srw1 downregulation prevents Nrm1 degradation in anaphase even as other substrates, such as Cdc13, are efficiently degraded. Nrm1 degradation motifs—a combination of one KEN box, a double DBox, and another non-standard motif that we termed CBox—are likely scattered across different regions of the protein. Any of the three motifs is sufficient to degrade Nrm1 through the proteasome Nrm1, since only the triple mutant results in a stable Nrm1 protein. Elucidating how these motifs contribute to Nrm1 recognition by the APC/C will require further investigation.

In summary, our work highlights the complex and highly regulated mechanisms by which fission yeast coordinates the G1/S transcriptional wave with cell cycle progression. The dual regulation of Nrm1 through phosphorylation and proteolysis underscores its pivotal role as both a repressor and an integrator of transcriptional and cell cycle signals. These findings also provide a framework for exploring conserved and divergent features of G1/S transcriptional regulation across eukaryotes.

## Methods

**Reagents and tools table**

| Reagent/resource | Reference or source | Identifier or catalog number |
|---|---|---|
| **Experimental models** | | |
| *E. coli* DH5a | N/A | N/A |
| *E. coli* BL21-DE3 | N/A | N/A |
| *Schizosaccharomyces pombe* strains | See Table EV1 | |
| **Recombinant DNA** | | |

| Reagent/resource | Reference or source | Identifier or catalog number |
|---|---|---|
| Plasmids used | See Table EV2 | |
| **Antibodies** | | |
| Anti-Cdc10 polyclonal | Gomez-Escoda 2010 | |
| Anti-GFP (JL8) | Takara | 632381 RID: AB_2313808 |
| Anti-HA monoclonal | Lab stock | 13CA5 |
| Anti-Myc | Lab Stock | 9E10 |
| Anti-Sty1 polyclonal | Jara 2007 | |
| Anti-thiophosphate ester | Abcam | Ab92570 |
| Anti-Tubulin | Sigma | T5168 RRID:AB_477579 |
| Anti-Yox1 | Ivanova 2013 | |
| Peroxidase Anti-Peroxidase Soluble Complex antibody (PAP) | Sigma | P1291 RRID: AB_1079562 |
| **Oligonucleotides and other sequence-based reagents** | | |
| Oligonucleotides used | See Table EV3 | |
| **Chemicals, enzymes and other reagents** | | |
| 1-NM-PP1 | TRC | A603003 |
| Ampicillin | Roche | 10 835 269 001 |
| Anhydrotetracyclin | Sigma-Aldrich | 37919-100MG-R |
| ATP | Promega | E6011 |
| ATPγS | Abcam | Ab138911 |
| Azide | Sigma | S8032 |
| Bradford | Bio-Rad | 5000006 |
| Calcofluor | Sigma | F3543 |
| Coomassie Brilliant Blue | Amresco | 0472 |
| Cycloheximide | Merk | C7698 |
| DNase I | Merk | 4716728001 |
| Formaldehide | Sigma | 252549 |
| High Capacity cDNA Reverse Transcription Kit with RNase inhibitor | Thermo Fisher | 4374966 |
| Hydroxyurea | Sigma-Aldrich | H8627-10G |
| Lambda phosphatase | New England Biolabs | P0753S |
| Lectin from glycine max soybean | Sigma | L1395 |
| LightCycler 480 Sybr Green I Master | Roche | 4887352001 |
| Methy methanesulfonate | Sigma-Aldrich | 129925 |
| $Ni^{2+}$-NTA agarose | Qiagen | 1018244 |
| Ni-NTA beads | Qiagen | 30210 |
| PhosStop | Roche | 4906845001 |
| PhosSTOP EASYpack | Roche | 4906845001 |

| Reagent/resource | Reference or source | Identifier or catalog number |
|---|---|---|
| P-Nitrobenzyl mesylate | Abcam | Ab138910 |
| Protease Inhibitor Cocktail | Sigma-Aldrich | P8215 |
| Protein Assay Dye Reagent Concentrate | BioRad | 5000006 |
| TCA | VWR | 1.00807.1000 |
| Thiamine | Merk | T1270 |
| **Software** | | |
| Gen5 | BioTek | |
| FACSDIVA software | BD Biosciences | |
| GraphPad Prism (8.0c) | GraphPad Software | https://www.graphpad.com/ |
| Python 3.9.19 | | https://www.python.org/downloads/release/python-390/ |
| R studio | | https://www.rstudio.com/ |
| **Other** | | |
| RNAseq data | This paper | GEO GSE287772 |

## Strains and media

All *S. pombe* strains used in this study (Table EV1) were constructed using standard yeast genetics. All plasmids used in this study are listed in Table EV2. Yeast media was prepared as described previously (Moreno et al, 1991). Ubc4 and Ubc11 point mutations were introduced by CRISPR (Torres-Garcia et al, 2020); plasmids and oligonucleotides used in CRISPR are listed in Tables EV2 and EV3, respectively. Cells were grown at 30 °C or at 25 °C for thermosensitive strains. HU 10 mM treatment was carried out on mid-log-grown cultures in YE5S media for 3 h. Tetracycline-induced promoters were induced overnight with 0.1 µg/ml anhydrotetracycline (Sigma-Aldrich, 37919-100MG-R) (Lyu et al, 2024). To analyze sensitivity to HU and MMS on plates, *S. pombe* strains were grown in liquid YE5S to an $OD_{600}$ of 0.5, serially diluted and plated onto YE5S media agar plates containing the indicated concentration of HU or MMS. Plates were incubated at 30 °C for 3–4 days.

## Synchronization experiments

Cells synchronicity was assayed by calcofluor staining as previously described (Moreno et al, 1991). For *cdc25-22* synchronization, cells were grown at 25 °C in Minimal Media until mid-log phase. Cultures were then shifted to 36 °C during 4 h and shifted back to 25 °C (Moreno et al, 1991). For *cdc2-asM17* synchronization, when cells reached mid-log phase the ATP analog 1NM-PP1 (Toronto Research Chemicals) was added to a final concentration of 1 µM. After 3 h, cells were filtered through a Millipore membrane, washed twice with fresh medium, and resuspended in Minimal Medium (Aoi et al, 2014). To arrest cells in metaphase, *nda3-KM311* cells were grown at 30 °C in YE5S until mid-log phase and shifted to 18 °C for 5 h (Baum et al, 1998).

## Cycloheximide-chase assays

Cells were grown to mid-log phase in permissive temperature. 100 µg/ml final concentration of cycloheximide was added to the cultures and thermosensitive strains were shifted to non-permissive temperature as indicated. For cells carrying the Nrm1 protein under the *nmt41* promoter, 2 µM of thiamine (final concentration) was also added.

## Preparation of *S. pombe* TCA extracts and immunoblot analysis

To analyze protein levels, trichloroacetic acid (TCA) extracts were prepared as previously described (Sanso et al, 2008). Immunoblot was performed after resolving the protein extracts on standard 8% PAGE, using monoclonal antibodies anti-HA (12CA5), polyclonal anti-Yox1 (Gómez-Escoda et al, 2011), monoclonal GFP JL-8 (Takara, 632381), anti-Myc (9E10) and anti-Cdc10 (Ivanova et al, 2013). For detection of Rep2-TAP Peroxidase Anti-Peroxidase Soluble Complex antibody (Sigma) was used. Anti-Sty1 (Jara et al, 2007) or anti-tubulin (Sigma, T5168) were used as loading control.

## Chromatin immunoprecipitation

Chromatin immunoprecipitation (ChIP) experiments were performed essentially as described previously (Moldon et al, 2008).

## RNA isolation and analysis by quantitative PCR

Cells were grown until mid-log phase and harvested by centrifugation at 2095 rcf for 3 min, washed with $H_2O$ and immediately frozen in ice. Each sample was then resuspended in 0.4 ml of AE buffer (50 mM sodium acetate, pH 5.3, 10 mM EDTA, pH 8.0). Sodium dodecyl sulfate was then added to a final concentration of 1%, and proteins and DNA were extracted by adding 0.4 ml of phenol-chloroform and incubated at 65 °C for 60 min vortexing every 15 min. The aqueous phase was separated by centrifugation at $10\,000 \times g$ for 5 min at 4 °C, extracted again in 0.4 ml of phenol-chloroform and before the final chloroform extraction. After chloroform extraction, RNA was precipitated with ethanol. 10 µg of purified RNA was treated with DNase I and reverse-transcribed to cDNA using Reverse Transcription System of Applied Biosystems (Thermo Fisher Scientific), following the manufacturer's instructions. cDNA was quantified by real-time quantitative PCR on Light Cycler II using Light Cycler 480 SYBR Green I Master (Roche). The error bars (standard deviation, SD) were calculated from at least three biological replicates, as indicated, and *tfb2* gene was used as a control for normalization. Fold induction was calculated comparing the value of each strain and condition to that of the asynchronous (AS) wild-type strain. Primers used are listed in Table EV3.

## RNAseq and analysis

Total RNA from JA3500, JA2176 and JA2179 strains was obtained as described above. Following ENCODE guidelines, two biological replicates were analyzed (RNA-seq couples have a Pearson correlation coefficient higher than 0.98). Libraries were prepared with NEBNext Ultra II Directional RNA Library Prep Kit for Illumina (New England Biolabs), with the polyA mRNA Magnetic Isolation Module, from total RNA. Libraries were then validated with the TapeStation (Agilent) and an equimolar pool was prepared and quantified by qPCR. The final pool was sequenced in a paired end run of $2 \times 75$ cycles High Output in a NextSeq500 (Illumina) instrument. For the RNAseq analysis, raw FASTQ files were first evaluated using quality control checks from FastQC 0.11.9. Alignment against the 'Schizosaccharomyces pombe all chromosomes 20230315' reference genome using STAR 2.7.9a. HTSeq was used for transcript quantification. Quantification results were then imported to DESeq2 1.36.0 for differential expression analysis. The threshold adjusted p-value used to identify differentially expressed genes was 0.1. Cluster analysis for differentially expressed genes was performed using DEGreport.

## Competition assays

Equal number of cells harboring mCherry/mRFP or sfGFP/eYFP was co-cultured in Minimal Medium, and diluted every 12 h to keep cultures in continuous exponential growth phase. Cells were kept at 4 °C in 50 mM NaCitrate pH 7.0. Cells were quantified using the LSRFortessa (BD Biosciences) cell analyzer. Data was analyzed with the FACSDIVA software (BD Biosciences).

## Pull-down of His$_6$-ubiquitin conjugates

To purify Nrm1-HA-His$_6$-ubiquitin conjugates, we made alkaline denaturing extracts of $3 \times 10^8$ cells. Briefly, cells were lysed in 8 M urea, 100 mM Tris-acetate pH 9.0, 100 mM Potassium acetate and 10 mM Magnesium acetate. Cells extracted were centrifuged at 4 °C in a microfuge for 5 min and the protein concentration was determined by Protein Assay Dye Reagent Concentrate (Bio-Rad). Four mg of extracted were mixed with $Ni^{2+}$-NTA agarose (Qiagen 1018244) and incubated overnight at 4 °C in a roller. The $Ni^{2+}$-NTA resin was washed three times in lysis buffer and resuspended in 1× Laemmli buffer. Samples were boiled for 5 min at 100 °C and analyzed by western blot with anti-HA antibodies.

## Immunoprecipitation of phosphorylated-Nrm1

To immunoprecipitate phosphorylated Nrm1, *nda3-KM311* cells were grown in YE5S at 30 °C until mid-log phase and then shifted to 18 °C during 5 h. Cells were lysed in NET-N (20 mM Tris-HCl pH 8.0, 100 mM NaCl, 1 mM EDTA, 0.5% NP-40) supplemented with protease inhibitors [0.5 mM PMSF, 1 mM DTT and Protease inhibitor cocktail (Sigma, P8215)] and Phosphatase inhibitor cocktail tablets PhosStop (Roche, 4906845001). Two mg of total extracts were immunoprecipitated with protein G crosslinked with anti-HA by incubation for 2 h at 4 °C in a roller. Immunoprecipitates were washed 3 times with 1 ml NET-N supplemented with protease and phosphatase inhibitors followed by 2 washes with 1X PBS. Beads were resuspended in PBS and treated with Lambda Protein Phosphatase (New England Biolabs, P0753S) in the presence and absence of PhosStop.

## Protein modelling with AlphaFold

Structures of Nrm1 A and Nrm1 C truncations interacting with Srw1 were predicted using Alphafold Server (Abramson et al, 2024). All 5 generated structures were energy optimized using

Rosetta relaxation protocol to minimize energy states (Conway et al, 2014). Models that retained favorable energetic and interaction profiles post-relaxation were selected for additional rounds of analysis and comparison with the original predicted models. Predicted models were further analyzed using Chimera X 1.9. The most significant side side-chain interactions were highlighted, specifically focusing on the contact regions of interest (KEN, DBox, CBox).

To model Nrm1 interaction with Cdc10, the whole MBF complex (2xCdc10, Res1, Res2, Rep2, 2xNrm1, 2xYox1) was introduced in the Alphafold Server. All 5 models generated were manually analyzed using Chimera X 1.9 in the search for contacts between Nrm1 and Cdc10 where a phosphorylatable Nrm1 residue faced a nearby negatively charged residue from Cdc10.

## Life-cell airyscan time-lapse experiments for cell cycle quantification

Precise cell cycle quantification for Nrm1, Nrm1-SA and Nrm1-SD was performed using the FLCCR strain as previously described (Murciano-Julia et al, 2025). Image acquisition was performed with a spectral confocal microscope Zeiss-LSM-982 with Airyscan 2 using a ×63 Plan-APO 1.46 NA objective. Image acquisition software used was ZEN 3.6.

Image analysis was performed using the TrackMate plugin (Tinevez et al, 2017) from Image J (Schindelin et al, 2012) as preciously described.

## His-tag protein purification from *E. coli*

BL21(DE3) *E. coli* cells containing the plasmid to express His-Nrm1 (pAY610) or His-Nrm1-SA (pAY1411) were grown in 200 ml of LB + 0.1 mg/ml ampicillin until $OD_{600} = 0.7$. Protein was induced o/n at 16 °C with 0.2 mM IPTG. Cells were harvested by centrifugation at 3800 rcf at 4 °C for 5 min. Pellet was resuspended in 5 ml of urea 8 M, 100 mM $NaH_2PO_4$ and 10 mM Tris-HCl at pH8. Cell lysis was performed by 5 pulses of 0.5 s of sonication at 40% amplitude. Soluble fraction was obtained by centrifugation at 12100 rcf at room temperature for 30 min. Then, 120 µl of $Ni^{2+}$-NTA agarose (Qiagen, 30210) was added to the soluble fraction and let rotate for 1 h at room temperature. The mix was loaded into an empty column (BioRad). The first flow-through was loaded once to ensure that the maximum amount of protein was bound to the beads. Beads were washed twice with 5 ml of urea 8 M, 100 mM $NaH_2PO_4$, 10 mM Tris-HCl at pH = 6.3. Protein was eluted from the beads with 250 µl of urea 8 M, 100 mM $NaH_2PO_4$, 10 mM Tris-HCl at pH = 3. Recombinant proteins were analyzed in a polyacrylamide gel and stained with Coomassie Brilliant Blue (Amresco, 0472). Purified proteins were stored at −80 °C after urea removal by dialysis against PBS.

## Non-radioactive in vitro kinase assay

Fifty ml of cells at $OD_{600} = 0.5$ were harvested by centrifugation at 2095 rcf for 3 min at 4 °C, transferred to screw cap tubes with 1 ml of cold PBS and pelleted at 16,100 rcf for 1 min at 4 °C. Pellet was then resuspended in NET-N buffer (20 mM Tris-HCl pH8, 100 mM NaCl, 1 mM EDTA, 0.5% NP-40) supplemented with protease inhibitors (PIC, Sigma-Aldrich P8215) and phosphatase inhibitors (PhosStop, Roche 4906845001) and cells were broken using a bead beater 3 cycles of 45 s. Debris was pelleted by centrifugation at 16100 rcf for 10 min and supernatant was recollected into a new tube. Protein concentration was quantified by Protein Assay Dye Reagent Concentrate (BioRad, 5000006). To immunoprecipitate Cdc2, 250 µg of protein extract were mixed with 20 µl of anti-HA beads and incubated at 4 °C for 2 h. Beads were washed by centrifugation at 4 rcf at 4 °C for 1 min twice with 500 µl in the breaking buffer (NET-N with protease and phosphatase inhibitors) and twice with 300 µl of kinase buffer 2X (100 mM HEPES pH7.5, 20 mM $MgCl_2$, 4 mM EGTA, 2 mM DTT). Each kinase reaction was done on the beads in a finale volume of 35 µl: 17.5 µl Kinase Buffer 2X, 0.7 µl ATP 1 mM (Promega E6011), 3.5 µl ATPγS (Abcam; ab138911) 10 mM and 3 µg of substrate. Kinase reaction was incubated at 30 °C for 20 min and stopped by adding 1.4 µl EDTA 0.5 M pH8. Then, 2 µl of 50 mM P-Nitrobenzyl mesylate (PNBM) (Abcam; ab138910) was added and incubated at 24 °C for 30 min. Esterification reaction was stopped by adding 5 µl SDS-PAGE sample buffer 5X and boiled for 5 min. Reactions were analyzed in two 10% SDS-PAGE. One gel for western blot detection using Rabbit anti-thiophosphate ester antibody (Abcam; ab92570) and anti-HA to detect the Cdc2 immunoprecipitated. The other gel was stained with Coomassie Brilliant Blue (Amresco, 0472) to check substrate loading.

## Growth curves

Growth curves were performed as previously described (Calvo et al, 2009). Briefly, exponentially growing cultures were diluted to an initial $OD_{600} = 0.1$ with the indicated treatments and were inoculated per duplicate in a 96-well non-coated polystyrene microplates with an adhesive seal. Plates were incubated in a Power Wave microplate scanning spectrophotometer (Bio-Tek) at 30 °C with continuous shaking. OD600 was automatically recorded every 10 min for 48 h using Gen5 software. Graphs were made using Pandas, Pyplot, and Seaborn packages from Python.

# Data availability

The data generated in the RNA-seq is available in GEO (GSE287772).

The source data of this paper are collected in the following database record: biostudies:S-SCDT-10_1038-S44319-025-00566-7.

# Peer review information

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

## Acknowledgements

We thank Kenji Kitamura (*slp1-362*), Takashi Toda (*skp1-A7*), Paul Nurse (*nda3-KM311*) and Masamitsu Sato (*cdc2-asM17*) for strains used in this work and the National BioResource Project (NBRP) for several plasmids and strains. We also thank Marta Isasa and Ross Tomaino for help with the Nrm1 phosphoproteomic results. We thank members of the Oxidative Stress and Cell Cycle Group for insightful comments and the staff from the CRG Advanced Light Microscopy Unit for invaluable technical help. This work was supported by grant PID2022-136449NB-I00 funded by MICIU/AEI/10.13039/501100011033, by Generalitat de Catalunya (Spain) (2021-SGR-00007) and ERDF/EU to JA and by Unidad de Excelencia Maria de Maeztu (CEX2018-000792-M) to JA and EH. EH is a recipient of an ICREA Academia Award (Generalitat de Catalunya). GM-J is a recipient of a FI predoctoral fellowship from Generalitat de Catalunya.

## Author contributions

**Guillem Murciano-Julià**: Conceptualization; Formal analysis; Investigation; Writing—original draft; Writing—review and editing. **Montserrat Vega**: Data curation; Writing—original draft. **Esther Pazo**: Investigation. **Àlex Pascual-Serra**: Investigation. **Isabel Alves-Rodrigues**: Investigation. **Oriol Bagudanch**: Investigation. **Roger Anglada**: Investigation. **Núria Bonet**: Investigation. **Rosa Aligué**: Conceptualization; Supervision. **Sergio Moreno**: Conceptualization; Resources. **Baldo Oliva**: Formal analysis; Supervision; Writing—original draft. **Elena Hidalgo**: Conceptualization; Resources; Supervision; Writing—original draft. **José Ayté**: Conceptualization; Resources; Formal analysis; Supervision; Funding acquisition; Writing—original draft; Writing—review and editing.

Source data underlying figure panels in this paper may have individual authorship assigned. Where available, figure panel/source data authorship is listed in the following database record: biostudies:S-SCDT-10_1038-S44319-025-00566-7.

## Disclosure and competing interests statement

The authors declare no competing interests.

# Expanded View Figures

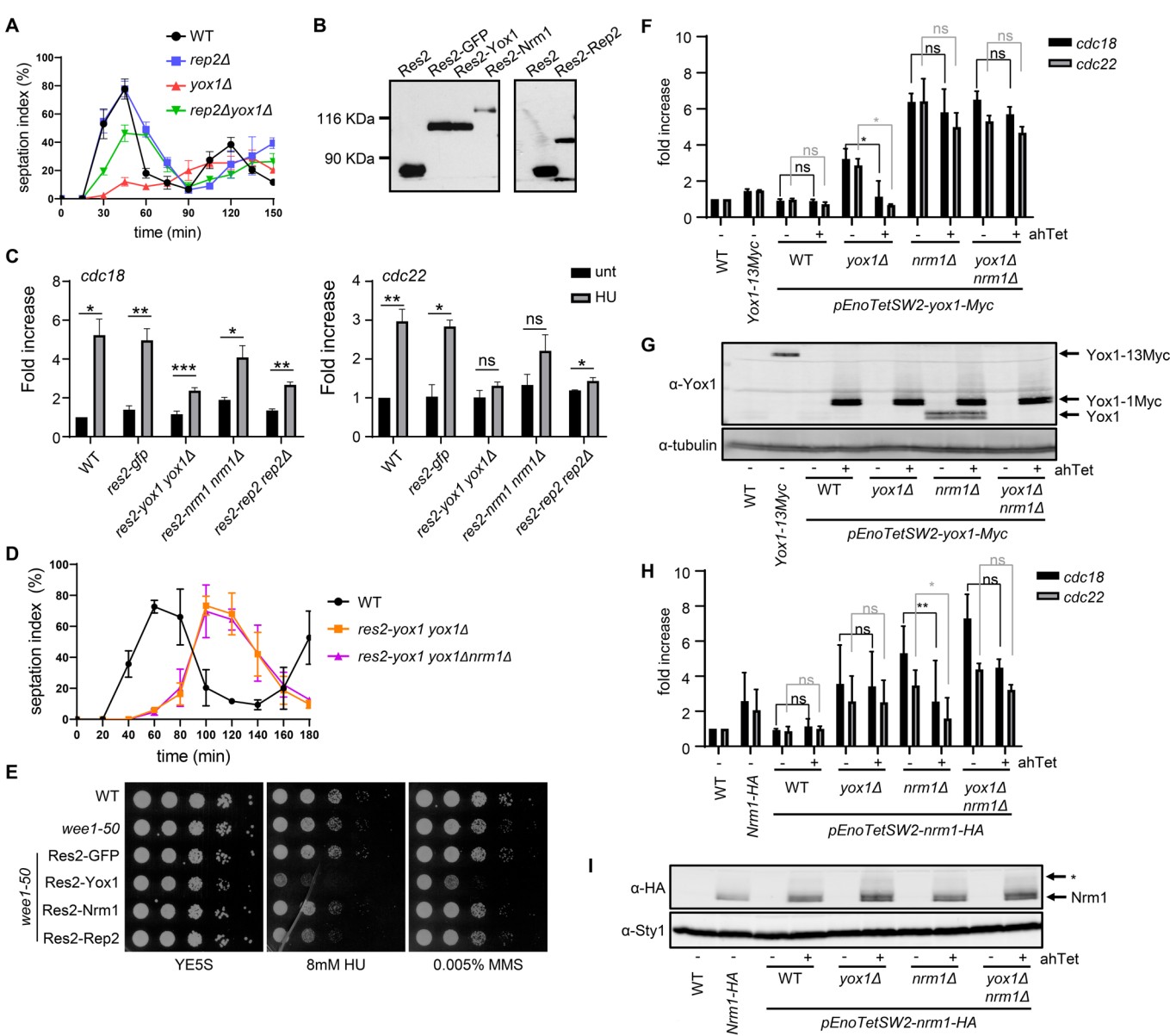

**Figure EV1.** Related to Fig. 1.

(A) Septation index of a wild type (WT, black), *rep2Δ* (blue), *yox1Δmik1Δ* (red) and *rep2Δyox1Δ* (green) in a *cdc2-asM17* block and release shown in Fig. 1B. Graphic represents mean ± SD of *n* = 3 experiments. (B) Western blot analysis of Res2 and Res2 chimeras. TCAs extracts were prepared from mid log cultures and separated in a 6% polyacrylamide gel. Western blot detection was performed with anti-HA antibodies. (C) qPCR of *cdc18* and *cdc22* expression in cultures grown in YE5S untreated (unt) or treated 3 h with 10 mM hydroxyurea (HU) of WT, *res2-gfp*, *res2-yox1 yox1Δ*, *res2-nrm1 nrm1Δ* and *res2-rep2 rep2Δ* cultures. *tfb2* was used as control gene. Expression was relativized to WT untreated. Graphic represents mean ± SD of *n* = 3 experiments. Statistics show significance from a Student's *T* test. ns: *p* > 0.05; **p* < 0.05; ***p* < 0.01; ****p* < 0.001. *p*(*cdc18* expression unt vs HU): 0.12325, 0.005623, 8.99E-06, 0.025819, 0.001346, respectively. *p*(*cdc22* expression unt vs HU): 0.008012, 0.020167, 0.179964, 0.142022, 0.026791. (D) Septation index of *cdc25-22* block and release experiment shown in Fig. 1D of WT (black), *res2-yox1 yox1Δ* (orange) *and res2-yox1 yox1Δnrm1Δ* (purple). Graphic represents mean ± SD of *n* = 3 experiments. (E) Survival was performed by spotting 10–105 cells of the indicated strains onto YE5S plates in the absence or presence of MMS or HU. Plates were incubated at 30 °C for 3–4 days. (F, G) qPCR of *cdc18* and *cdc22* (F) and α-Yox1 western blot (G) of WT, *yox1-13Myc* and WT, *yox1Δ*, *nrm1Δ* and *yox1Δnrm1Δ* strains expressing Yox1-1Myc under the control of a tetracycline (ahTet) inducible promoter *pEnoTetSW2*. *tfb2* was used as control gene in qPCRs and expression was then relativized to WT non-induced sample. Graphic represents mean ± SD of *n* = 3 experiments. Statistics show significance from a Student's *T* test. ns: *p* > 0.05; **p* < 0.05; ***p* < 0.01; ****p* < 0.001. *p*(*cdc18* expression unt vs ahTet): 0.9174, 0.0139, 0.5449 and 0.1358, respectively. *p*(*cdc22* expression unt vs ahTet): 0.0932, 0.0093, 0.03241 and 0.0929, respectively. In (G), it is shown one representative experiment out of three different experiments. α-Yox1 was used to detect endogenous and overexpressed Yox1. Tubulin is shown as a loading control. (H, I) Same experiment as in (F, G), but cells were expressing Nrm1-3HA under the control of a tetracycline (ahTet) inducible promoter *pEnoTetSW2*. In (H), *tfb2* was used as control gene in qPCRs and expression was then relativized to WT non-induced sample. Graphic represents mean ± SD of *n* = 3 experiments. Statistics show significance from a Student's *T* test. ns: *p* > 0.05; **p* < 0.05; ***p* < 0.01; ****p* < 0.001. *p*(*cdc18* expression unt vs ahTet): 0.5425, 0.9250, 0.0087 and 0.1104, respectively. *p*(*cdc22* expression unt vs ahTet): 0.6741, 0.9343, 0.0236 and 0.1170, respectively. In (I), one representative experiment is shown out of three different. α-HA was used to detect Nrm1-3HA expression. Sty1 is shown as a loading control.

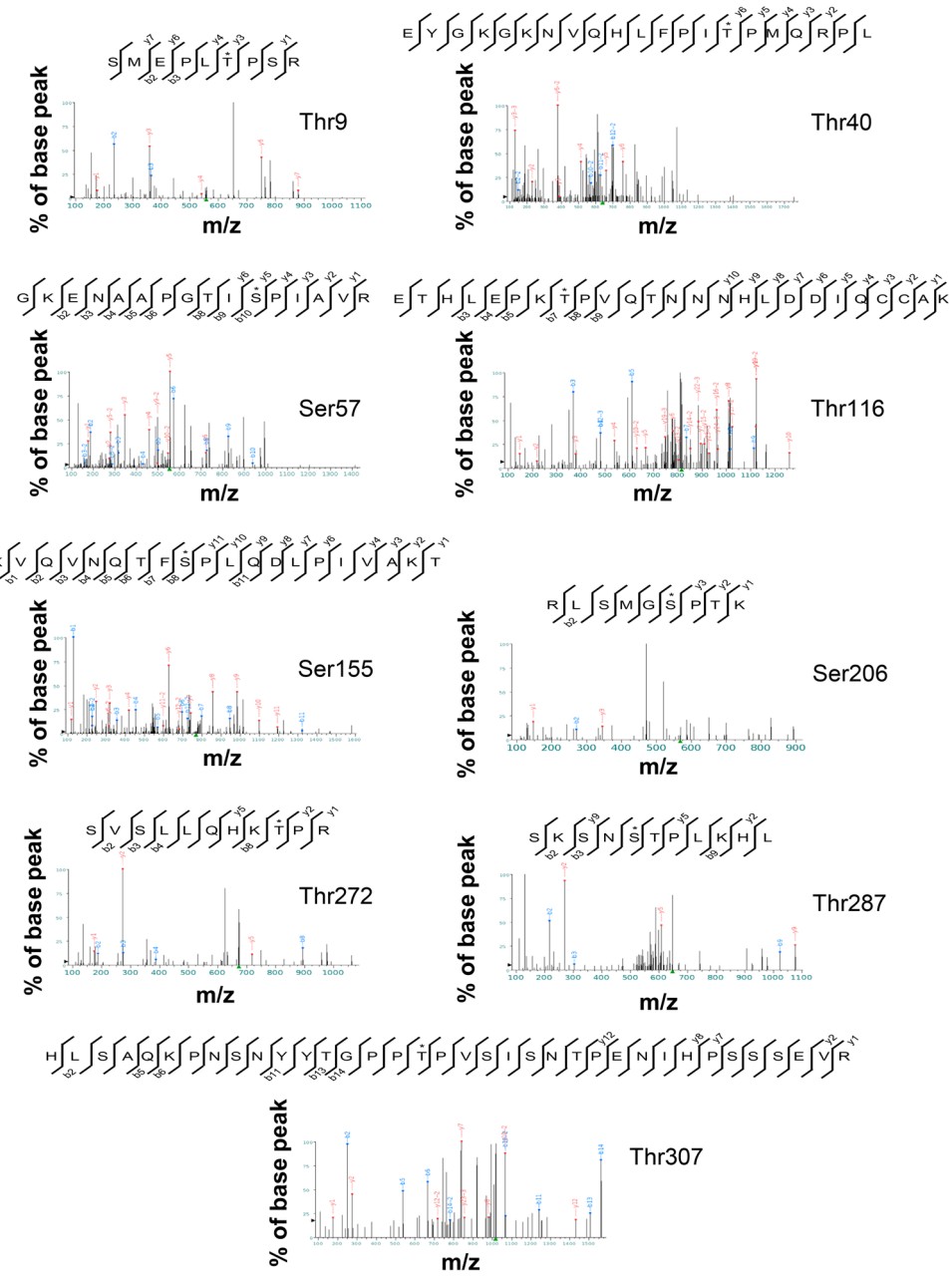

**Figure EV2. Related to Fig. 3.**

Original mass spectra map of putative phosphorylation sites in Nrm1. Purified Nrm1-HA from metaphase arrested cells was trypsin and/or chymotrypsin digested. The resulting peptides were analyzed by LC-MS/MS. The collision-induced dissociation MS2 and sequence of the phosphorylated peptides are shown.

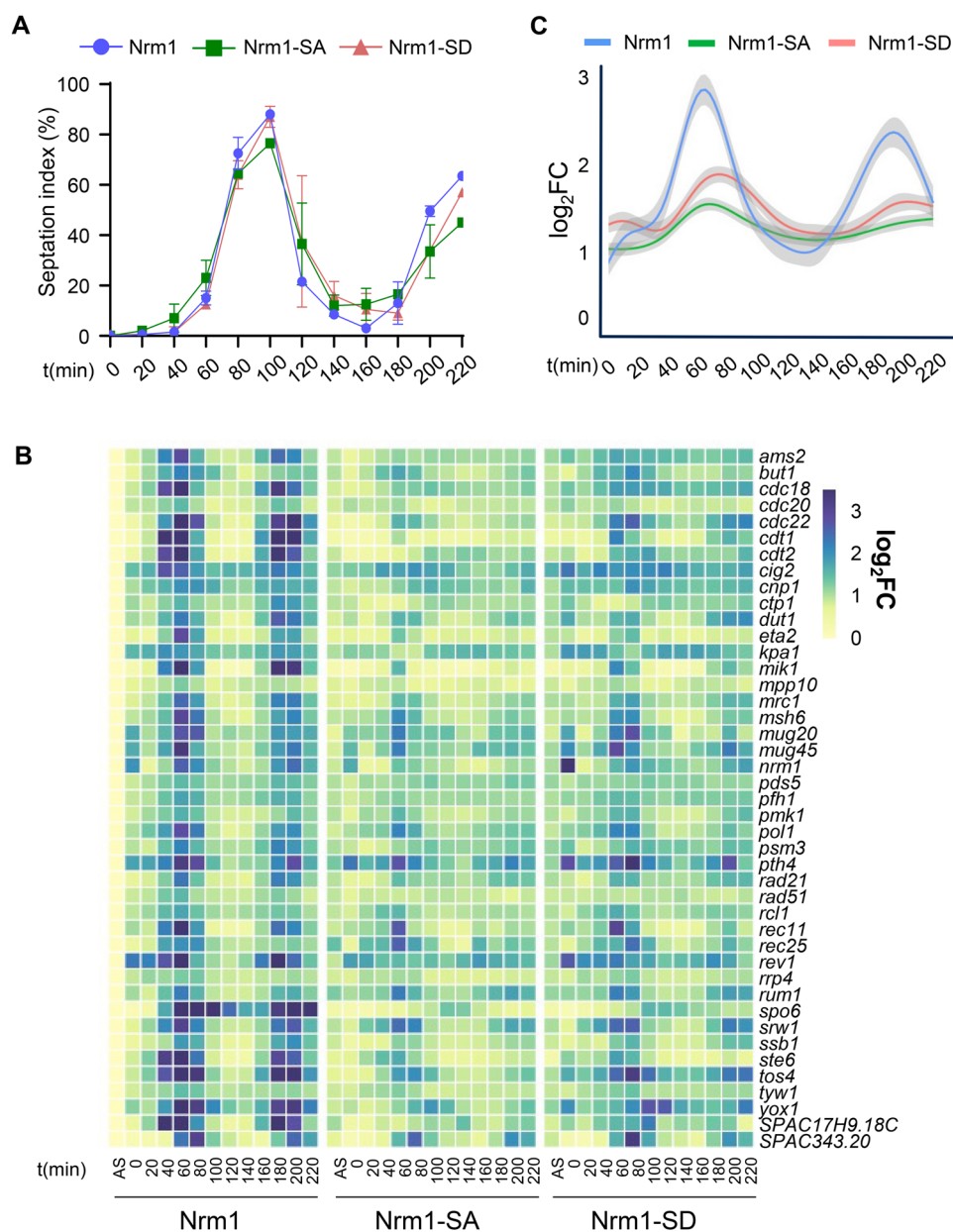

**Figure EV3.  Related to Fig. 4.**

(A) Septation index of wild type (Nrm1, blue), Nrm1-SA (green) and Nrm1-SD (red) in a *cdc25-22* block and release shown in Fig. 4C,D. Graphic represents mean ± SD of $n = 3$ experiments. (B) Heatmap of the gene expression of 43 MBF-dependent genes. RNAseq of wild type (Nrm1), Nrm1-SA and Nrm1-SD in a *cdc25-22* block and release. Genes were selected as cycling genes with peaks of transcription at 60–80 and 180–200 min after release and having expression altered in Nrm1-SA and Nrm1 SD strains. Scale represents Log₂FC of normalized reads of each time-point relative to Nrm1 asynchronous (AS) values. (C) Expression of MBF-dependent genes from Fig. EV3B in a wild type (Nrm1, blue), Nrm1-SA (green) and Nrm1-SD (red). Line represents mean of values and grey shadow represents SD.

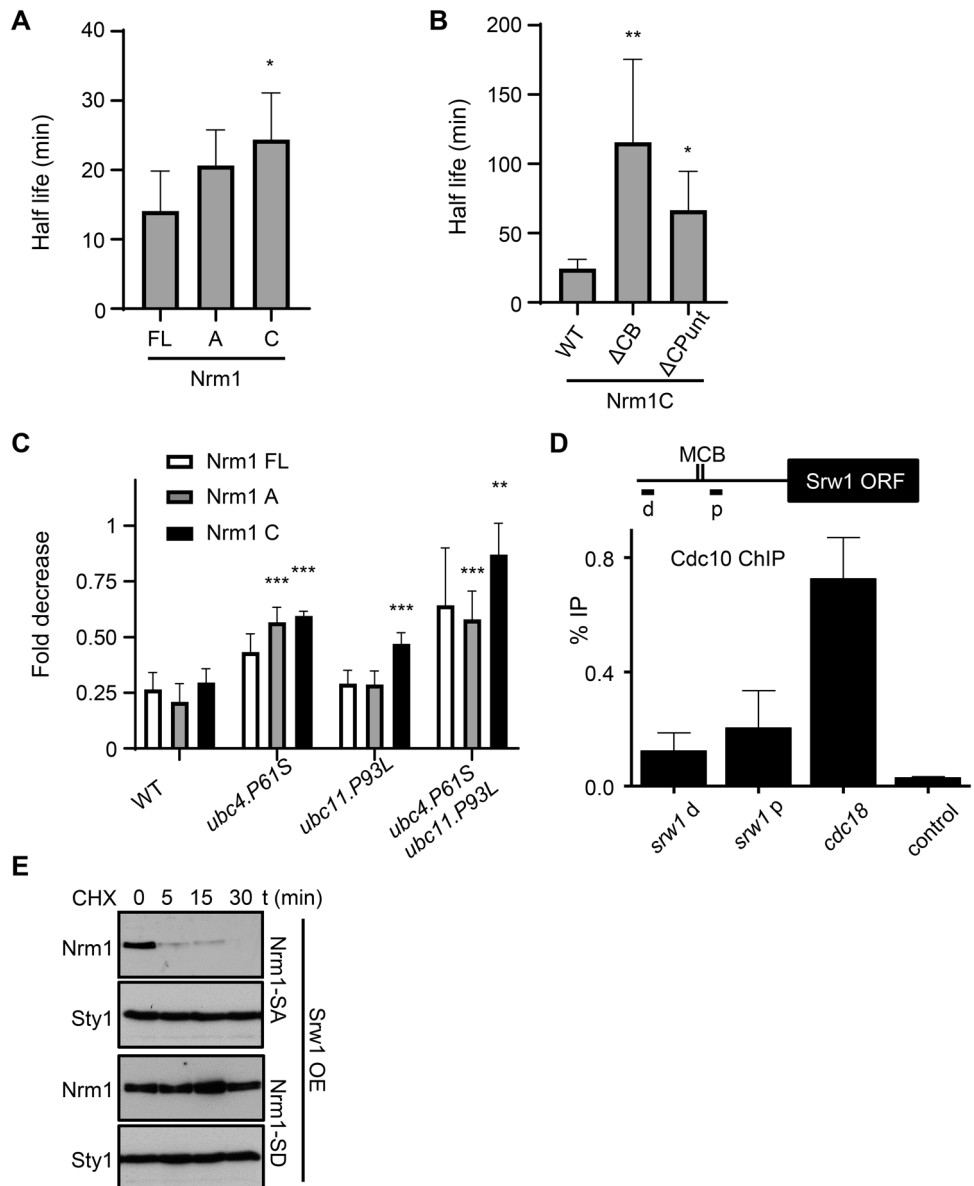

**Figure EV4. Related to Fig. 5.**

(A) Half-life in minutes of wild type full length Nrm1 (FL), the first half of Nrm1 containing the first 171 residues (A) and the carboxi-domain of Nrm1 containing the last 171 residues (C). TCA samples were collected at 0, 10, 30 and 60 min after cycloheximide addition and analyzed in western blot. Sample loading was normalized with Sty1. Half-life was calculated using GraphPad Prism. Plot represents mean ± SD of at least $n = 3$ experiments. Statistics show significance from a Student's $T$ test. *$p < 0.05$; **$p < 0.01$; ***$p < 0.001$. p(Nrm1 FL vs Nrm1 A): 0.0641. p(Nrm1 FL vs Nrm1 C): 0.0232. (B) Half-life in minutes of the carboxi-domain of Nrm1 containing the last 171 residues (Nrm1 C) native sequence (WT), or with a deletion of the CBox (ΔCB) or with 5 amino acid substitutions (H252A W255A, R263A, V265A, L267A) (ΔCPunt). TCA samples were collected at 0, 10, 30 and 60 min after cycloheximide addition and analyzed in western blot. Sample loading was normalized with Sty1. Half-life was calculated using GraphPad Prism. Plot represents mean ± SD of at least $n = 3$ experiments. Statistics show significance from a Student's $T$ test. *$p < 0.05$; **$p < 0.01$; ***$p < 0.001$. p(Nrm1 C WT vs ΔCB): 0.0096. p(Nrm1 C WT vs ΔCB): 0.0149. (C) Protein fold change (FC) of Nrm1 full length (Nrm1-FL), Nrm1-A and Nrm1-C in wild type (WT), ubc4.P61S, ubc11.P93L, and ubc4.P61S ubc11.P93L strains. Cells were grown at 25 °C and shifted at 37 °C for 1 h before adding cycloheximide. Samples were collected before and 1 h after the treatment. TCA extracts were performed and assayed by western blot. Loading was normalized with Sty1. Plot represents mean ± SD of at least $n = 3$ experiments. Statistics show significance from a Student's $T$ test. *$p < 0.05$; **$p < 0.01$; ***$p < 0.001$. p(Nrm1 FL vs WT): 0.06, 0.666 and 0.0717, respectively. p(Nrm1 A vs WT): 2.44E-05, 0.1107 and 0.0004, respectively. p(Nrm1 C vs WT): 3.9E-05, 0.0064 and 0.0001, respectively. (D) Cdc10 Chip from asynchronous cultures; srw1 d, distal region of srw1 promoter; srw1 p, proximal region of srw1 promoter. On top, a scheme of srw1, indicating the 2 MCB sites in the promoter; d, distal amplicon; p, proximal amplicon. Plot represents mean ± SD of at least $n = 3$ experiments. Statistics show significance from a Student's $T$ test. *$p < 0.05$; **$p < 0.01$; ***$p < 0.001$. (E) Western blot of extracts prepared from wild type cells overexpressing Srw1 (Srw1 OE). Cells were expressing endogenous Nrm1-SA or Nrm1-SD. On top, time after the addition of cycloheximide (CHX). Sty1 is shown as loading control. One representative experiment is shown out of three different experiments.

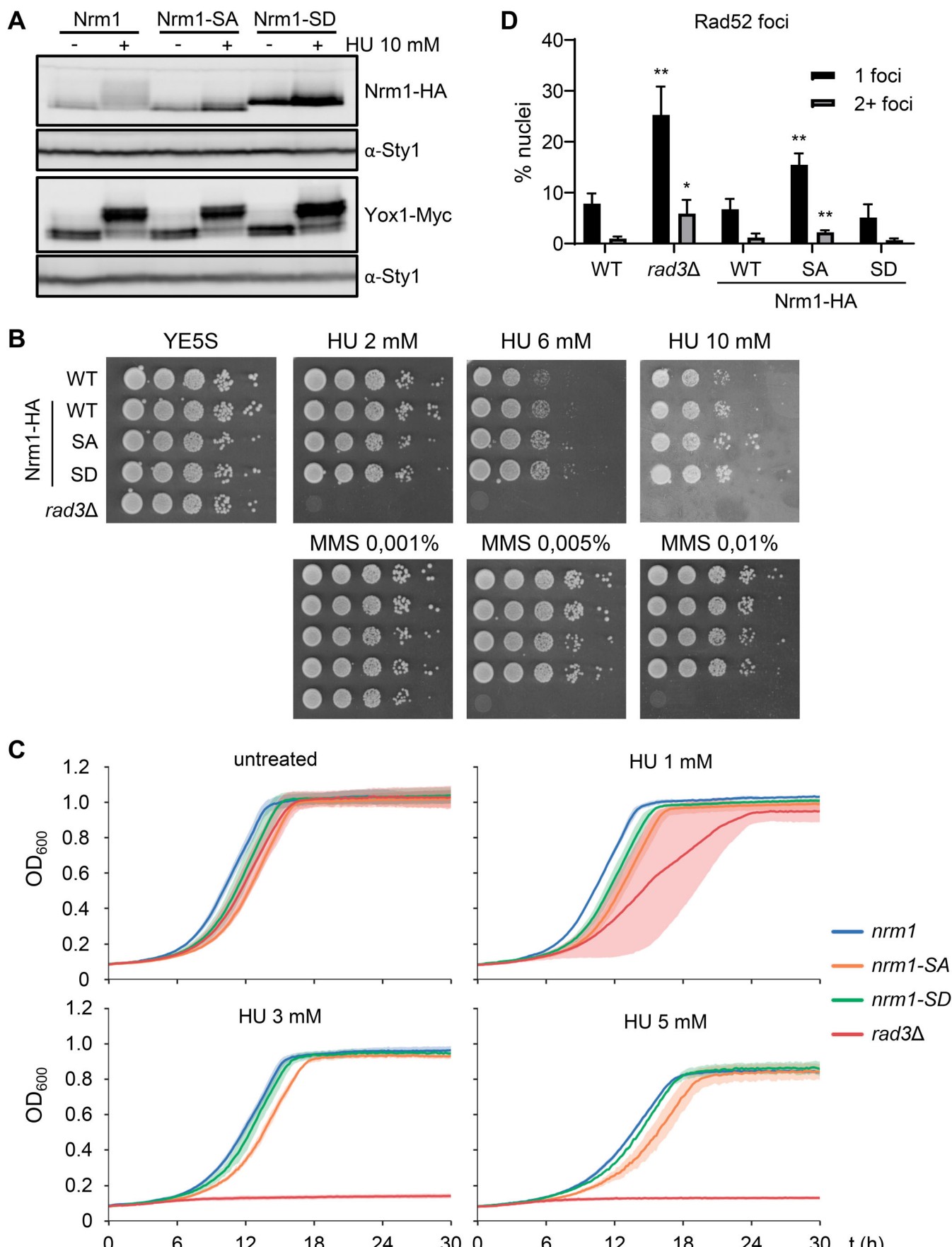

**Figure EV5.** Related to Fig. 6.

(A) Western blot analysis of Nrm1 and Yox1 in strains with wild type Nrm1, Nrm1-SA or Nrm1-SD. Cells were collected before and after treatment with hydroxyurea (HU) 10 mM for 3 h. Sty1 is shown as loading control. One representative experiment is shown out of three different replicas. (B) Survival was performed by spotting $10–10^5$ cells of the indicated strains onto YE5S plates in the absence or presence of the indicated concentrations of MMS or HU. Plates were incubated at 30 °C for 3–4 days. (C) Growth curves comparing growth at 30 °C of the indicated strains in YE5S in the absence or presence of different concentrations of hydroxyurea (HU). Plot represents mean ± SD of at least $n = 3$ experiments. (D) Percentage of nuclei containing Rad52-mNeonGreen foci in WT, *rad3Δ, nrm1-HA, nrm1-SA-HA* and *nrm1-SD-HA* strains grown in YE5S. At least 100 nuclei were counted per replica. Plot represents mean ± SD of at least $n = 3$ experiments. Statistics show significance from a Student's *T* test. *$p < 0.05$; **$p < 0.01$. $p$(1 foci vs WT): 0.0010, 0.4701, 0.0051 and 0.1745, respectively. $p$(2+ foci vs WT): 0.0121, 0.6422, 0.0099 and 0.3624, respectively.

