## [Peer Review File · EMBO Reports]

Nrm1 is a bistable switch connecting cell cycle progression to transcriptional control

Guillem Murciano-Julià, Montserrat Vega, Esther Pazo, Alex Pascual-Serra, Isabel Alves-Rodrigues, Oriol Bagudanch, Roger Anglada, Núria Bonet, Rosa Aligue, Sergio Moreno, Baldo Oliva, Elena Hidalgo, and José Ayté

Corresponding author: José Ayté (jose.ayte@upf.edu)

Review Timeline:

Submission Date:	24th Feb 25
Editorial Decision:	21st Mar 25
Revision Received:	16th Jun 25
Editorial Decision:	23rd Jul 25
Revision Received:	23rd Jul 25
Accepted:	11th Aug 25

Editor: Deniz Senyilmaz Tiebe

Transaction Report:

Dear Prof. Ayté,

Thank you for submitting your research manuscript to our journal, which was now seen by three referees, whose reports are copied below.

Referees express interest in the proposed mechanism by which Nrm1 connects cell cycle progression to transcriptional control. However, they also raise some concerns that need to be addressed to consider publication here.

I find the reports informed and constructive, and believe that addressing the concerns raised will significantly strengthen the manuscript. As the reports are below, and I think all points need to be addressed, I will not detail them here. Please contact me if you have questions or comments regarding the revision for further discussion (also by video chat).

Given these recommendations, we would like to invite you to submit a revised manuscript. Please revise your manuscript with the understanding that the referee concerns (as in their reports) must be fully addressed and their suggestions taken on board. Please address all referee concerns in a complete point-by-point response. Acceptance of the manuscript will depend on a positive outcome of a second round of review. It is EMBO reports policy to allow a single round of major experimental revision only and acceptance or rejection of the manuscript will therefore depend on the completeness of your responses included in the next, final version of the manuscript.

We realize that it is difficult to revise to a specific deadline. In the interest of protecting the conceptual advance provided by the work, we recommend a revision within 3 months. Please discuss the revision progress ahead of this time with me if you require more time to complete the revisions, or if you have questions or comments regarding the revision (also by video chat).

1. A data availability section providing access to data deposited in public databases is missing (where applicable).
2. Your manuscript contains statistics and error bars based on $n=2$. Please use scatter plots in these cases.

You can submit the revision either as a Scientific Report or as a Research Article. For Scientific Reports, the revised manuscript can contain up to 5 main figures and 5 Expanded View figures, and it should not exceed 27000 characters. If the revision leads to a manuscript with more than 5 main figures it will be published as a Research Article. In this case the Results and Discussion section should be separate. If a Scientific Report is submitted, these sections have to be combined. This will help to shorten the manuscript text by eliminating some redundancy that is inevitable when discussing the same experiments twice. In either case, all materials and methods should be included in the main manuscript file.

4) a .docx formatted letter INCLUDING the reviewers' reports and your detailed point-by-point responses to their comments. As part of the EMBO publication's Transparent Editorial Process, EMBO reports publishes online a Review Process File (RPF) to accompany accepted manuscripts. This File will be published in conjunction with your paper and will include the referee reports,

your point-by-point response and all pertinent correspondence relating to the manuscript.

<https://www.embopress.org/page/journal/14693178/authorguide#transparentprocess>

5) a complete author checklist, which you can download from our author guidelines

<https://www.embopress.org/page/journal/14693178/authorguide>. Please insert information in the checklist that is also reflected in the manuscript. The completed author checklist will also be part of the RPF.

6) Please note that all corresponding authors are required to supply an ORCID ID for their name upon submission of a revised manuscript (<<https://orcid.org/>>). Please find instructions on how to link your ORCID ID to your account in our manuscript tracking system in our Author guidelines

<<https://www.embopress.org/page/journal/14693178/authorguide#authorshipguidelines>>

7) Before submitting your revision, primary datasets produced in this study need to be deposited in an appropriate public database (see <https://www.embopress.org/page/journal/14693178/authorguide#datadeposition>). Please remember to provide a reviewer password if the datasets are not yet public. The accession numbers and database should be listed in a formal "Data Availability" section placed after Materials & Method (see also

<https://www.embopress.org/page/journal/14693178/authorguide#datadeposition>). Please note that the Data Availability Section is restricted to new primary data that are part of this study. * Note - All links should resolve to a page where the data can be accessed. *

Additional information on source data and instruction on how to label the files are available:

<https://www.embopress.org/page/journal/14693178/authorguide#sourcedata>

9) Our journal encourages inclusion of *data citations in the reference list* to directly cite datasets that were re-used and obtained from public databases. Data citations in the article text are distinct from normal bibliographical citations and should directly link to the database records from which the data can be accessed. In the main text, data citations are formatted as follows: "Data ref: Smith et al, 2001" or "Data ref: NCBI Sequence Read Archive PRJNA342805, 2017". In the Reference list, data citations must be labeled with "[DATASET]". A data reference must provide the database name, accession number/identifiers and a resolvable link to the landing page from which the data can be accessed at the end of the reference. Further instructions are available at <http://www.embopress.org/page/journal/14693178/authorguide#referencesformat>

10) Regarding data quantification (see Figure Legends:

<https://www.embopress.org/page/journal/14693178/authorguide#figureformat>)

11) The journal requires a statement specifying whether or not authors have competing interests (defined as all potential or actual interests that could be perceived to influence the presentation or interpretation of an article). In case of competing interests, this must be specified in your disclosure statement. Further information: <https://www.embopress.org/competing->

interests

12) Please also note our reference format:

13) All Materials and Methods need to be described in the main text using our 'Structured Methods' format, which is required for all research articles. According to this format, the Methods section includes a Reagents and Tools Table (listing key reagents, experimental models, software and relevant equipment and including their sources and relevant identifiers) followed by a Methods and Protocols section describing the methods using a step-by-step protocol format. The aim is to facilitate adoption of the methodologies across labs. More information on how to adhere to this format as well as a downloadable template (.docx) for the Reagents and Tools Table can be found in our author guidelines:

I look forward to seeing a revised version of your manuscript when it is ready. Please let me know if you have questions or comments regarding the revision.

Kind regards,

Deniz Senyilmaz Tiebe

Deniz Senyilmaz Tiebe, PhD
Senior Scientific Editor
EMBO Reports

Referee #1:

Murciano-Julià et al. present thorough and insightful analysis of the regulation of G1/S transcriptional regulation in fission yeast. Much is known about the factors involved, but they and their regulation have generally been studied in the context of checkpoint regulation. Murciano-Julià et al. work out the regulatory mechanisms in an unperturbed cell cycle. They find that Nrm1 phosphorylation and degradation are key steps in interconnected feedback loops that activate, and then inactivate, G1/S transcription. The work is well done and well presented. It will be of broad interest in the fields of cell cycle and transcription, both for the specific about fission yeast regulation and for the interesting and important comparisons to analogous regulation in budding yeast and metazoans. Nonetheless, the manuscript would be improved by attention to the following minor points.

On Page 5, the statement "As shown in Fig. 1A, deletion of *yox1* (or *nrm1*)" indicates that *nrm1*Δ data is included in Figure 1, which it is not.

The experiment shown in Figure EV1C should be repeated and the results quantitated.

The *cdc2-asM17* experiments described on Page 7 demonstrate that Cdc2 is required for Nrm1 phosphorylation, but not that it directly phosphorylates Nrm1. The phrase "the kinase responsible for Nrm1 phosphorylation" indicates the later, and thus should be modified.

It is expected that Nrm1-SD would not be phosphorylated. Starting the last sentence of the first paragraph on Page 8 implies otherwise, and should be modified.

Nrm1-SD has much more repressive function than a null allele and presumably more than fully phosphorylated Nrm1. So, referring to it as "constitutively OFF" on Page 12 is misleading.

On Page 14, when referring to the proteins involved in G1/S in "mammalian cells and budding yeast", (RB) should be (RB/Whi5) and (E2F) should be (E2F/MBF,SBF).

Referee #2:

Murciano-Julia et al. investigate the cell-cycle dependent regulation of the MBF transcription factor. They show that Nrm1 is a substrate of the mitotic CDK, and CDK1-mediated phosphorylation leads to release of Nrm1 from chromatin, thus coupling the

G1/S transcription wave to cell-cycle progression. They also show an additional layer of regulation, which involves the APC-dependent degradation of Nrm1, regulated both by phosphorylation by CDK1, and by an MBF target, Srw1, in a feedback loop. The technical quality of the experiments is generally high and their wide range of approaches makes full use of the fission yeast model system. The conclusions are largely supported by the data, with some inaccuracies (see minor comments).

Major comment:

They show in co-culturing experiment that the Nrm1-SA and Nrm1-SD mutants have diminished cellular fitness, even though the impact on cell-cycle progression (Fig 6D and E) is relatively minor. It would be interesting to explore the specifics of this loss of fitness, and the consequences of the mutations on the fate of the cells. Such experiments would greatly add general interest and relevance to the field in general, as well as enhance the discussion of parallels and differences in the regulation of the G1/S transcriptional wave in human cells and in budding yeast. I am suggesting addressing questions like: do the SA and SD cells suffer a higher rate of replication stress? Are they dependent on checkpoint function in an unperturbed cell cycle? Are they more vulnerable to DNA damage? Is phosphorylation on the CDK1 sites required for checkpoint-induced MBF activation (when CDK activity is inhibited)?

Minor comments:

P 5 "yox1Δ cells exhibited constitutive activation of MBF-dependent transcription throughout the entire cell cycle, losing the characteristic peak at the G1/S transition" Fig EV1A the septation index in the yox1Δ cells shows very poor synchronization, which could be the reason for the apparent loss of cell-cycle regulation of MBF activity.

P 6 I find the description of results on the fusion proteins is somewhat confusing. First, it looks to me that the cells expressing Res2-Nrm1 CAN induce cdc18, but not as much as the controls (Fig EV1C). Furthermore, the expression levels of the fusion proteins are different, in particular Res2-Nrm1 is expressed to much lower levels than Res2 or Res2-Yox1. It would be interesting to see the expression levels of the endogenous Yox1 and Nrm1 if there are suitable antibodies. Could overexpression of the repressors lead to dominant phenotypes?

Second, the statement that Res2-Yox1 cells cannot induce cdc18 or cdc22 is not quite supported by the data presented. There IS a little induction of cdc22, not so much of cdc18, so the overall conclusion is correct, but the statement is too strong. The septation index shown in EV1D indicates a delay in the res2-yox1 cells, which is not explained. However, it means that the labelling of the cell-cycle phases under x in Fig 1D is not correct.

Fig2B It is not clear whether the statements regarding cell-cycle regulation of protein levels are based on one or several experiments. Quantification of protein levels would strengthen the case.

Fig 2C the mobility shift is very difficult to see - at least on my screen. A longer exposure should also be shown.

Fig 4A As above for Fig 2B; does the figure show a representative blot of several experiments? One experiment? A quantification of the band intensities (upper / total) would strengthen the claim.

General comments on the bandshifts: it is not always easy to detect phosphorylation as such clear bandshifts. Here they present very nice immunoblots of large bandshifts, which either show very heavy phosphorylation - or high levels of immunoblotting skills and maybe a lot of optimization. A comment in the Methods should be made to say whether the blots shown are from phostag gels, any particular percentage or type of gel, or maybe standard methods, only very large differences in mobility.

Referee #3:

In this manuscript, the authors propose a two-step mechanism underlying the MBF cell cycle-dependent activation through the release of the corepressor Nrm1 in fission yeast cells. Major improvements should be made to the manuscript to make it suitable for consideration for publication in the journal EMBO Reports. Specifically, the authors should address the following concerns:

- 1) While the study identifies CDK1 as the kinase responsible for Nrm1 phosphorylation during the proposed two-step mechanism of activating MBF-dependent transcription, the manuscript lacks detailed mechanistic insights into how this phosphorylation event is regulated and its exact impact on Nrm1's interaction with the MBF complex. Further experiments, such as more directed mutagenesis, could provide a deeper understanding of the proposed regulatory mechanism for Nrm1 as a bistable switch connecting the cell cycle with transcription.
- 2) In line with the above, the authors claim that CDK1-mediated phosphorylation of Nrm1 triggers its release from chromatin in metaphase. However, while they show that phosphorylation correlates with Nrm1 release and MBF activation, they do not directly demonstrate a causal relationship. For instance, in vitro phosphorylation assays or the use of phospho-specific antibodies could strengthen this claim.
- 3) I would like to commend the clever approach of fusing either an activator or repressor directly to the MBF component. This same system could be used to test the requirement of Nrm1 for Yox1 repression. Repeating the experiment shown in Figure 1D in an Nrm1 deletion background would strengthen the authors' claims.
- 4) In Figures 3C and 3D, the authors claim differences in expression and binding when using WT, SA, or SD versions of Nrm1. However, the statistical analysis is currently only performed between differently synchronized cells. The authors should directly compare the WT, SA, and SD Nrm1 versions in the same synchronized cells to strengthen their claims.
- 5) The ChIP experiments shown in Figure 2A and 3D require additional control using an MBF-independent promoter region to draw conclusive results.
- 6) For all AlphaFold prediction experiments, the authors should report the predicted confidence levels of the models, particularly

the Predicted Aligned Error (PAE) information between the predicted interacting regions. This information is essential for readers to assess the reliability and informativeness of the predictions.

7) In Figure 5I, the authors should show the full structures of the proteins and the positions of all identified degradation motifs. Currently, the structure appears to depict only partial segments of the proteins, making it impossible to assess where the different degradation motifs are localized.

8) In the RNA-seq experiment shown in Figure 4B, the authors demonstrate that *nrm1* and *yox1* are also MBF target genes and that their expression is altered when expressing Nrm1-SA or Nrm1-SD. The authors should provide evidence that the effects observed with Nrm1-SA/SD are not due to changes in Yox1 levels. At the very least, they should discuss the potential feedback loops present in the cells.

9) The study suggests a feedback loop involving Srw1 and Nrm1. While the data imply that Srw1 promotes the degradation of Nrm1, the mechanistic details remain somewhat speculative. Also, the data presented in Figure 5H is not entirely convincing. The Nrm1-HA levels in the OE negative samples should be equal. Currently, the variability between the signals appears to be high, requiring additional biological replicates to convincingly demonstrate the effect of Srw1. For example, direct assays showing Srw1-dependent ubiquitination of Nrm1 (e.g., in vitro ubiquitination assays) would provide more definitive evidence.

10) While the study identifies multiple degradation motifs in Nrm1, it would be valuable to determine the individual contributions of each motif to Nrm1 degradation. Experiments involving single motif mutants (e.g., KEN box alone, DBox alone, and CBox alone) could help assess their specific roles in Nrm1 stability. Furthermore, comparing the degradation rates of these single mutants to the triple mutant could clarify the redundancy and specificity of these motifs in the context of Nrm1 regulation.

11) While the manuscript references relevant literature, it would benefit from a more comprehensive discussion of how this study fits into the broader context of cell cycle regulation and transcriptional control. Emphasizing the novel aspects of this work in relation to previous studies would strengthen the manuscript.

Minor comments

1) The current manuscript title might be confusing. It is not immediately evident what the authors mean by the word "bistable" in the title.

2) The authors might want to rephrase the following sentence in the abstract: "This mechanism overlaps with the irreversible degradation of unphosphorylated Nrm1 (which originates from de novo synthesis or by dephosphorylation of pre-existing Nrm1 during anaphase), preventing its re-association with MBF until the end of the following S phase.", as the degradation of proteins is always irreversible.

3) The authors should explain in more detail how ChIP was carried out in the Methods section. Also, what is the "control" region used in the experiment shown in Fig EV4D?

4) Please write all centrifugations as rcf, not rpm to ensure reproducibility.

5) Please verify that the genome assembly versions used for RNA-seq data alignment is correct. Right now, the used name "Schizosaccharomyces pombe all chromosomes 20230315" does not seem to correspond to official assembly versions.

6) Please add the software name used to assign reads to features in RNA-seq experiment.

7) Page 19 sentence "Cells were kept at 4°C un 50 mM NaCitrate pH7" - should read "in", not "un".

8) Page 21, title "Life-cell airyscan time-lapse experiments for cell cycle quantification" should possibly be "live-cell."

9) Figure 2E - some phosphorylation still remains after inhibiting CDK1. Thus, the claim of "abolishes phosphorylation" might need to be softened.

10) Figure 3A - the figure would benefit from adding the color code information as a legend in the figure itself. Also, depicting where the Cbox is located would be beneficial.

11) The authors should clearly define what the dCB and dCpnt mutations are. Currently, the reader has to assume that the dCB mutation involves replacing all ten residues with alanine, while the dCpnt mutation involves changing five amino acids as indicated in the text. However, the text in the Results section (page 10) states: "To evaluate the contribution of these domains to Nrm1 stability, we introduced mutations in these motifs-either replacing all ten residues with alanines or introducing point mutations in two residues of each motif-and measured the half-life of the resulting mutants." This implies that the dCpnt mutation involves four mutations, not five, which needs clarification.

Referee #1:

Murciano-Julià et al. present thorough and insightful analysis of the regulation of G1/S transcriptional regulation in fission yeast. Much is known about the factors involved, but they and their regulation have generally been studied in the context of checkpoint regulation. Murciano-Julià et al. work out the regulatory mechanisms in an unperturbed cell cycle. They find that Nrm1 phosphorylation and degradation are key steps in interconnected feedback loops that activate, and then inactivate, G1/S transcription. The work is well done and well presented. It will be of broad interest in the fields of cell cycle and transcription, both for the specific about fission yeast regulation and for the interesting and important comparisons to analogous regulation in budding yeast and metazoans. Nonetheless, the manuscript would be improved by attention to the following minor points

We would like to thank the encouraging comments from the reviewer. As she/he will observe, we have answered all her/his comments.

On Page 5, the statement "As shown in Fig. 1A, deletion of *yox1* (or *nrm1*)" indicates that *nrm1* Δ data is included in Figure 1, which it is not.

We have now included the data also from the *nrm1* Δ strain in Fig 1A.

The experiment shown in Figure EV1C should be repeated and the results quantitated. We have repeated the experiment (n=3) and replaced the old Northern blots with Q-PCR, which facilitates the quantifications:

The *cdc2*-asM17 experiments described on Page 7 demonstrate that Cdc2 is required for Nrm1 phosphorylation, but not that it directly phosphorylates Nrm1. The phrase "the kinase responsible for Nrm1 phosphorylation" indicates the later, and thus should be modified.

Following the reviewer advice, we have changed the text to: "...implicating that CDK1 is directly or indirectly involved in Nrm1 phosphorylation". Also, it is worth to mention that as a request from another reviewer, we have also now included an in vitro kinase assay, showing that an IP of Cdc2-HA is able to phosphorylate Nrm1 (and not Nrm1-SA), which reinforces the idea that CDK1 indeed phosphorylates Nrm1 (new Figure 3C).

It is expected that Nrm1-SD would not be phosphorylated. Starting the last sentence of the first paragraph on Page 8 implies otherwise, and should be modified.

Thanks. We have changed the text to "*nrm1-SD cells, in which it was also abolished the metaphase mobility shift, had a different basal mobility compared to the wild type cells (Fig. 3B, last 3 lanes), probably due to the increased negative charge of the Nrm1-SD when compared to unphosphorylated wild type Nrm1.*"

Nrm1-SD is has much more repressive function than a null allele and presumably more than fully phosphorylated Nrm1. So, referring to it as "constitutively OFF" on Page 12 is misleading.

The reviewer is absolutely right; this sentence could be misleading. We have changed the text to: "... *further emphasizing that fixing Nrm1 phosphorylation status -whether constitutively phosphorylated or unphosphorylated- results in equivalent fitness defects.*"

On Page 14, when referring to the proteins involved in G1/S in "mammalian cells and budding yeast", (RB) should be (RB/Whi5) and (E2F) should be (E2F/MBF,SBF).

We have changed the text to: "... *activation occurs through phosphorylation of the repressor (RB or Whi5, respectively), which promotes its release from the transcription factor (E2F or SBF, respectively).*"

Referee #2:

Murciano-Julia et al investigate the cell-cycle dependent regulation of the MBF transcription factor. They show that Nrm1 is a substrate of the mitotic CDK, and CDK1-mediated phosphorylation leads to release of Nrm1 from chromatin, thus coupling the G1/S transcription wave to cell-cycle progression. They also show an additional layer of regulation, which involves the APC-dependent degradation of Nrm1, regulated both by phosphorylation by CDK1, and by an MBF target, Srw1, in a feedback loop. The technical quality of the experiments is generally high and their wide range of approaches makes full use of the fission yeast model system. The conclusions are largely supported by the data, with some inaccuracies (see minor comments).

We would like to thank the encouraging comments from the reviewer. As she/he will observe, we have tried to follow all her/his recommendations which, as we believe, have noticeably improved the manuscript.

Major comment:

They show in co-culturing experiment that the Nrm1-SA and Nrm1-SD mutants have diminished cellular fitness, even though the impact on cell-cycle progression (Fig 6D and E) is relatively minor. It would be interesting to explore the specifics of this loss of fitness, and the consequences of the mutations on the fate of the cells. Such experiments would greatly add general interest and relevance to the field in general, as well as enhance the discussion of parallels and differences in the regulation of the G1/S transcriptional wave in human cells and in budding yeast. I am suggesting addressing questions like: do the SA and SD cells suffer a higher rate of replication stress? Are they dependent on checkpoint function in an unperturbed cell cycle? Are they more vulnerable to DNA damage? Is phosphorylation on the CDK1 sites required for checkpoint-induced MBF activation (when CDK activity is inhibited)?

Following the reviewer comment, we have tested both strains (*nrm1-SA* and *nrm1-SD*) for replicative stress and DNA damage, checking for Yox1 phosphorylation and Rad52 foci in unperturbed cells, growth curves in the presence of low concentrations of HU, sensitivity to HU and MMS in plates. All these experiments are shown in Fig. EV5. As the reviewer can observe, neither *nrm1-SA* nor *nrm1-SD* cells have induced the replication stress (Yox1 is not constitutively phosphorylated) and they are not sensitive to HU or MMS on solid plates. They have a very subtle sensitivity that can be observed on growth curves at mild HU concentrations, where Nrm1-SA is more sensitive than Nrm1-SD, which is more sensitive than a WT strain (EV5D). Finally, when measuring DNA damage (Rad52 foci, EV5B), we observe an increase of Rad52 foci in the Nrm1-SA strain. We have also included the corresponding text commenting on the characteristics of both mutants.

Regarding the last point raised by the reviewer (*Is phosphorylation on the CDK1 sites required for checkpoint-induced MBF activation?*), we show in Fig 3C that induction of *cdc18* and *cdc22* after HU treatment is similar in WT and *nrm1-SA* or *nrm1-SD* strains; this happens because Yox1 is still phosphorylated by Cds1 after HU-treatment and it is released from the MBF complex (Gomez-Escoda et al (2011), EMBO Rep), inducing MBF-dependent transcription, independently of Nrm1 phosphorylation status. All in all, the new experiments shown in Fig EV5 reinforce the idea that CDK-dependent phosphorylation of Nrm1 is critical to regulate MBF-dependent transcription in coordination with cell cycle progression, but are mostly unrelated to DNA replication stress (which is mediated through phosphorylation of Yox1, Nrm1 and Cdc10 by Cds1 at different sites).

Minor comments:

P 5 "yox1Δ cells exhibited constitutive activation of MBF-dependent transcription throughout the entire cell cycle, losing the characteristic peak at the G1/S transition" Fig EV1A the septation index in the yox1Δ cells shows very poor synchronization,

which could be the reason for the apparent loss of cell-cycle regulation of MBF activity. Indeed, *yox1Δ* cells are tough to synchronize in a *cdc25-22* or in a *cdc2-as* background. One of the reasons for this could be that they are constitutively expressing *mik1*; so, a *cdc2-as yox1Δ mik1Δ* has improved synchronization, as it is shown in Figure 1B and Fig EV1A.

Alternatively, we have done the synchronization experiments in a *cdc2-25 yox1Δ mik1Δ*, which has better synchronization, as can be seen here:

As the reviewer can observe, synchronization in a *cdc25-22 yox1Δ mik1Δ* is improved, when compared with a *cdc2-as* synchronization (although it is still much worse than in the wild type *cdc25-22*); and the transcription of *cdc18* does not change along the whole experiment, because *cdc18* (and that of *cdc22*) is constitutive in the absence of Yox1, like in the experiment that we show in Fig 1B (which is done in a *cdc2-as* background).

So, one possibility to overcome this problem would be to repeat the experiment shown in Figure 1B using the *cdc25-22* allele to synchronize the cultures; the problem is that *cdc25-22 rep2Δ* and specially *cdc25-22 res1Δ* cells are extremely sick and cannot be used in synchronization experiments.

In any case, the key point we wish to convey to the reviewer -and to future readers of the manuscript, if accepted- is that regardless of the quality of synchronization, the absence of Yox1 leads to complete de-repression of MBF-dependent gene transcription. This conclusion is further supported by Figure 1A, which shows that in the absence of either Yox1 or Nrm1, MBF-dependent transcription is robustly induced.

P 6 I find the description of results on the fusion proteins is somewhat confusing. First, it looks to me that the cells expressing Res2-Nrm1 CAN induce *cdc18*, but not as much as the controls (Fig EV1C). Furthermore, the expression levels of the fusion proteins are different, in particular Res2-Nrm1 is expressed to much lower levels than Res2 or Res2-Yox1. It would be interesting to see the expression levels of the endogenous Yox1 and Nrm1 if there are suitable antibodies. Could overexpression of the repressors

lead to dominant phenotypes?

First of all, we have repeated the experiment with the fusion proteins as shown in Fig EV1C (triplicates of Q-PCR experiments instead of the Northern blot that was shown originally).

Indeed, *cdc18* and *cdc22* are induced in the Res-Nrm1 after HU treatment. The explanation that we find is that since upon replicative stress (+HU) Yox1 is phosphorylated inducing its release from MBF and MBF-dependent transcription (as described in Gomez-Escoda, EMBO Rep, 2011). This activation of MBF is reduced in the Res2-Yox1 quimera, since Yox1 is covalently bound to MBF and maintains the repression.

Regarding the different amount of the fusion proteins, all them are expressed fused to Res2 at the *res2* loci and under the control of the *res2* promoter. What happens is that while GFP, Yox1 and Rep2 are stable proteins, Nrm1 has an extremely short half-life explaining why there is less amount of this fusion than of the rest fused proteins. To note, in these strains, there are no endogenous copies of the proteins that were fused to Res2; that is, in the Res2-Yox1 there is no endogenous Yox1 (strain JA1202: *h+yox1Δ::ura4+ res2-yox1-Nat+ ura4-D18*); in the Res2-Nrm1 there is no endogenous Nrm1 (strain JA1203: *h-nrm1Δ::kan+ res2-nrm1-Nat+*); in the Res2-Rep2 there is no endogenous Rep2 (strain JA2033: *h-rep2Δ::KanMX6 res2-linker-rep2-3HA-NatMX6*). Regarding the last point raised by the reviewer on the effect of overexpression of Yox1 or Nrm1, the effect depends on the genetic background, but they do not lead to dominant phenotypes (see plots below). Overexpression of Nrm1, does not have any effect on a WT strain, and cannot repress the transcription of MBF-dependent genes in a $\Delta yox1$ strain or in a $\Delta yox1 \Delta nrm1$ strain; however, it can restore normal levels of MBF transcription on a $\Delta nrm1$ strain. Regarding overexpression of Yox1, MBF-dependent transcription is unaffected in all the strains except in the $\Delta yox1$ strain (same as it happens with the overexpression of Nrm1). The results related to overexpression of Nrm1 or Yox1 have been added to Fig. EV1

Second, the statement that Res2-Yox1 cells cannot induce *cdc18* or *cdc22* is not quite supported by the data presented. There IS a little induction of *cdc22*, not so much of *cdc18*, so the overall conclusion is correct, but the statement is too strong. The septation index shown in EV1D indicates a delay in the *res2-yox1* cells, which is not explained. However, it means that the labelling of the cell-cycle phases under x in Fig 1D is not correct.

The reviewer is absolutely right. We have changed the text softening our conclusion that Res2-Yox1 cells cannot induce *cdc22* or *cdc18*: "On the contrary, cells expressing a Res2-Yox1 chimera showed a diminished capability to induce MBF-dependent genes after HU treatment, especially in the case of *cdc22*. In fact, Res2-Yox1 cells that had by-passed the function of Nrm1 since it was no longer required for loading Yox1 onto

MBF, where unable to induce cdc18 or cdc22 during the G1/S transition either in cells with or without Nrm1 (Fig. 1D and Fig. EV1D)”.

We have also changed that labelling of the graph in Fig 1D and added a sentence pointing to the fact that res2-yox1 cells have delayed G1/S transition. We have also included the expression of cdc18 and cdc22 in a parallel block & release experiment of cells expressing the Res2-Yox1 chimera in a $\Delta nrm1$ background. We have also added a sentence indicating that septation index is delayed in the res2-yox1 cells: *“which correlates with a delayed septation peak in synchronous cultures when compared to wild type cultures, reflecting problems with the initiation of DNA replication (Fig. EV1D)”*

Fig2B It is not clear whether the statements regarding cell-cycle regulation of protein levels are based on one or several experiments. Quantification of protein levels would strengthen the case.

We have repeated the experiments several times ($n \geq 3$), but we are just showing a representative one. Some of these experiments were done many years ago, so we are not convinced that we would be able to quantitate all them in the same way (some are chemiluminiscent WB; other are developed with immunofluorescent secondary antibody). And the main take home message for the reader would be that Nrm1 is not present during all the cell cycle, while the other components are present during all the cell cycle (we do not discuss regarding small changes on the amount of the proteins along the cell cycle).

We have changed the text to indicate that this is a representative WB of at least 3 different experiments: *“Representative Western Blots of at least 3 different experiments are shown.”*

Fig 2C the mobility shift is very difficult to see - at least on my screen. A longer exposure should also be shown.

We believe that the reviewer means Fig2B, Nrm1 panel; we have added an image with higher contrast corresponding to the WB of Nrm1

Fig 4A As above for Fig 2B; does the figure show a representative blot of several experiments? One experiment? A quantification of the band intensities (upper / total) would strengthen the claim.

This is a representative experiment of an $n=3$. We have changed the text to indicate that this is a representative WB of at least 3 different experiments: *“Representative Western Blots of at least 3 different experiments are shown.”*

Furthermore, like in Figure 2B, the only aim of this figure is to show that Nrm1-SA and Nrm1-SD are stable, contrary to what happens to WT Nrm1

General comments on the bandshifts: it is not always easy to detect phosphorylation as such clear bandshifts. Here they present very nice immunoblots of large bandshifts, which either show very heavy phosphorylation - or high levels of immunoblotting skills and maybe a lot of optimization. A comment in the Methods should be made to say whether the blots shown are from phostag gels, any particular percentage or type of gel, or maybe standard methods, only very large differences in mobility.

No, they are no PhosTag gels; they are regular Laemli 8% PAGE gels. We believe that since Nrm1 is phosphorylated in so many residues, we were lucky to observe clear band shifts upon phosphorylation. A sentence has been added to the Methods section: *“Immunoblot was performed after resolving the protein extracts on standard 8% PAGE”*

Referee #3:

In this manuscript, the authors propose a two-step mechanism underlying the MBF cell cycle-dependent activation through the release of the corepressor Nrm1 in fission yeast cells. Major improvements should be made to the manuscript to make it suitable for consideration for publication in the journal EMBO Reports. Specifically, the authors should address the following concerns:

We would like to thank the reviewer for his/her insightful comments. As she/he will discern, we have diligently endeavored to incorporate all of their recommendations, which, in our estimation, have significantly enhanced the manuscript.

1) While the study identifies CDK1 as the kinase responsible for Nrm1 phosphorylation during the proposed two-step mechanism of activating MBF-dependent transcription, the manuscript lacks detailed mechanistic insights into how this phosphorylation event is regulated and its exact impact on Nrm1's interaction with the MBF complex. Further experiments, such as more directed mutagenesis, could provide a deeper understanding of the proposed regulatory mechanism for Nrm1 as a bistable switch connecting the cell cycle with transcription.

In the laboratory, we have generated several mutants of Nrm1 at the CDK phosphosites (S/T-P; in bold, strict CDK consensus sites, S/T-P-X-K/R), including:

Nrm1-3A: Nrm1-**T9A** S57A **T287A** (JA1785)

Nrm1-4A: Nrm1-**T9A** S57A T116A **T287A** (JA1786)

Nrm1-5A: Nrm1-**T9A** S57A **S237A T241 T287A** (JA1787)

Nrm1-6A: Nrm1-**T9A** S57A T116A **S237A T241A T287A** (JA1792)

We tested these mutants for mobility shift upon mitotic arrest (left panels, WBs) and the effect on *cdc22* transcription (right panel).

As the reviewer can observe, not even Nrm1-6A mutant can preclude the mobility shift on mitotically arrested cells and *cdc22* is still induced in mitosis over 2-fold. We did not include these (and more experiments with other mutants), since the only form to completely abolish the mobility shift and *cdc22* transcription induction was using the Nrm1-11A mutant. Now, we have added these figures to the Appendix PDF section of the manuscript which, as proposed by the reviewer, may add a deeper understanding of the mechanism regulating Nrm1 and MBF-dependent transcription.

2) In line with the above, the authors claim that CDK1-mediated phosphorylation of Nrm1 triggers its release from chromatin in metaphase. However, while they show that phosphorylation correlates with Nrm1 release and MBF activation, they do not directly demonstrate a causal relationship. For instance, in vitro phosphorylation assays or the use of phospho-specific antibodies could strengthen this claim.

We have now done in vitro kinase assays, showing that IPed Cdc2-HA can phosphorylate Nrm1, but not Nrm1-SA. We have included this experiment in Figure 3C.

3) I would like to commend the clever approach of fusing either an activator or repressor directly to the MBF component. This same system could be used to test the

requirement of Nrm1 for Yox1 repression. Repeating the experiment shown in Figure 1D in an Nrm1 deletion background would strengthen the authors' claims.

The experiments shown in Figure 1D and all the chimeras used in this manuscript were done in absence of other source of the fused proteins. That is, the Res2-Yox1 strain was generated in a $\Delta yox1$ background. The same applies to the Res2-Nrm1 and Res2-Rep2 strains. Following the suggestion from the reviewer, we have repeated the experiment of the Res2-Yox1 chimera in a $\Delta yox1\Delta nrm1$ background. As the reviewer can observe, the effect of Nrm1 is minimal in a strain in which Yox1 is covalently linked to MBF. We have included a comment of this observation in our manuscript: *"In fact, Res2-Yox1 cells that had by-passed the function of Nrm1 since it was no longer required for loading Yox1 onto MBF, where unable to induce cdc18 or cdc22 during the G1/S transition either in cells with or without Nrm1 (Fig. 1D and Fig. EV1D), pointing that the primary repressor could be Yox1"*

We have also changed Figure EV1D to include the septation index of the $\Delta yox1\Delta nrm1$ strain.

4) In Figures 3C and 3D, the authors claim differences in expression and binding when using WT, SA, or SD versions of Nrm1. However, the statistical analysis is currently only performed between differently synchronized cells. The authors should directly compare the WT, SA, and SD Nrm1 versions in the same synchronized cells to strengthen their claims.

The idea behind these 2 figures (3C and 3D) was to show the different behavior of Nrm1, Nrm1-SA and Nrm1-SD in mitosis (initiation of MBF activation) when compared to the asynchronous cultures of the same strain; and, importantly, to show that the response under replicative stress (+HU) was mostly unaffected in the mutants (since it is dependent of different sites being phosphorylated) when compared to the WT strain. We have now included a version of Figs. 3C and 3D in the Appendix PDF section of the manuscript that includes the statistical analysis requested by the reviewer:

Stats with WT
Stats with SA

And also, here are the t-test values among all the values:

t-test among all values																			
cdc18										cdc22									
	AS	HU	M	AS	HU	M	AS	HU	M	AS	HU	M	AS	HU	M	AS	HU	M	
AS	1									AS	1								
HU	0,007	1								HU	0,007	1							
M	0,003	0,307	1							M	0,003	0,059	1						
AS	0,770	0,005	0,002	1						AS	0,169	0,004	0,002	1					
HU	0,006	0,107	0,240	0,003	1					HU	0,087	0,972	0,150	0,061	1				
M	0,601	0,005	0,001	0,737	0,003	1				M	0,313	0,022	0,006	0,093	0,148	1			
AS	0,123	0,009	0,003	0,022	0,007	0,017	1			AS	0,146	0,013	0,005	0,019	0,139	0,982	1		
HU	0,006	0,441	0,801	0,004	0,243	0,004	0,009	1		HU	0,016	0,932	0,068	0,009	0,986	0,041	0,031	1	
M	0,556	0,012	0,009	0,398	0,022	0,323	0,594	0,014	1	M	0,073	0,055	0,010	0,025	0,257	0,349	0,269	0,097	1

We hope that we have convinced the reviewer that with these 2 figures we show that they behave differently at the onset of MBF activation (mitotic arrested cells), but not in cells under replicative stress.

5) The ChIP experiments shown in Figure 2A and 3D require additional control using an MBF-independent promoter region to draw conclusive results.

We use an intergenic region or mitochondrial DNA in all our ChIP experiments, but this was not included in our graphs for graphic simplicity. We have included the information in the Methods section.

Figure 2A CHIP + INTERGENIC CONTROLS

Figure 3D CHIP + INTERGENIC CONTROLS

We have now added this graphs in the Appendix PDF section of the manuscript.

6) For all AlphaFold prediction experiments, the authors should report the predicted confidence levels of the models, particularly the Predicted Aligned Error (PAE) information between the predicted interacting regions. This information is essential for readers to assess the reliability and informativeness of the predictions.

We have included AlphaFold predictions not as a way to infer possible interactions between different proteins, but to show that our experimental findings may correlate with possible structural predictions. For example, in Fig 3F (PAE shown below), we hypothesize that a possible phosphorylation of T9 could force an electrostatic repulsion with E764 in Cdc10, which could explain why Nrm1 phosphorylation at T9 could induce the release of Nrm1 from the MBF complex. To determine if this prediction is true, it would require to generate compensatory mutations on Cdc10 (for example E764R). These are experiments we do not want to get involved with, since it would be beyond the scope of this manuscript. We have also changed the text to indicate that these are just basic AlphaFold predictions that could coincide with and justify our experimental results, and that they are not intended to generate experiments.

7) In Figure 5I, the authors should show the full structures of the proteins and the positions of all identified degradation motifs. Currently, the structure appears to depict only partial segments of the proteins, making it impossible to assess where the different degradation motifs are localized.

The complete MBF complex contains 2 molecules of Cdc10, 1 Res1, 1 Res2, 2 Yox1, 2 Nrm1 and 1 Rep2; the MWR of the complex is 463 KDa. We have not found any way to show with clarity the predicted structure of such a large complex (see image below). This is why we decided to show partial segments, aligning Nrm1 and Srw1 and the potential recognition motifs found in our manuscript. Again, as indicated in the previous concern of the reviewer, we do not want to generate new data based on hypothetical AlphaFold prediction structures, but to show that there are theoretical predictions that align with our experimental results. We have changed the text accordingly. Below, please have a view of the prediction of the whole MBF complex using AlphaFold:

The color code is as follows: A, B: Cdc10 (grey); C: Res1 (orange); D: Res2 (orange-red); E: Rep2 (pink); F, G: Nrm1 (blue); H, I: Yox1 (green); J, K: DNA (blue and red)

8) In the RNA-seq experiment shown in Figure 4B, the authors demonstrate that *nrm1* and *yox1* are also MBF target genes and that their expression is altered when expressing Nrm1-SA or Nrm1-SD. The authors should provide evidence that the effects observed with Nrm1-SA/SD are not due to changes in Yox1 levels. At the very least, they should discuss the potential feedback loops present in the cells.

We agree with the reviewer: regulation of MBF is very complex, with multiple feedback regulation. Not only Yox1 and Nrm1 are negative feedback regulators, but Cig2 is also a negative feedback regulator (Ayte et al, 2001, Nat Cell Biol); and we show now that Srw1/Ste9 is a positive feedback regulator, which directly impinge onto a negative regulator, Nrm1. On top of this complex loops, protein localization also is important: Nrm1-SD is (at least partially) excluded from chromatin, which affects Nrm1 loading on MBF and it is unable to repress MBF-dependent transcription. Furthermore, we are now showing that overexpression of Nrm1 or Yox1 does not have any measured effect on MBF-dependent transcription (new panels in Fig EV1), which reinforces one of the ideas that flow from our manuscript: it is not the quantity of Nrm1, but the phosphorylation status of Nrm1 which regulate MBF activity. We have included some text in the manuscript pointing to this, as suggested by the reviewer.

9) The study suggests a feedback loop involving Srw1 and Nrm1. While the data imply that Srw1 promotes the degradation of Nrm1, the mechanistic details remain somewhat speculative. Also, the data presented in Figure 5H is not entirely convincing. The Nrm1-HA levels in the OE negative samples should be equal. Currently, the variability between the signals appears to be high, requiring additional biological replicates to convincingly demonstrate the effect of Srw1. For example, direct assays showing Srw1-dependent ubiquitination of Nrm1 (e.g., *in vitro* ubiquitination assays) would provide more definitive evidence.

We agree with the reviewer that the WB shown in Fig. 5H was not the best one. We have done several times the experiments overexpressing Slp1, Skp1 and Srw1 (see below), and are replacing the old figure with a new one. And we also show in Fig EV4E that overexpression of Srw1 induces Nrm1 degradation when cannot be phosphorylated (Nrm1-SA).

Regarding the suggestion of the reviewer to include an *in vitro* ubiquitination assay, there is no doubt that it will be definitely to determine which E3 subunit(s) is (are) participating in the degradation of Nrm1; however, we find that it could be part of future work of the laboratory.

10) While the study identifies multiple degradation motifs in Nrm1, it would be valuable to determine the individual contributions of each motif to Nrm1 degradation. Experiments involving single motif mutants (e.g., KEN box alone, DBox alone, and CBox alone) could help assess their specific roles in Nrm1 stability. Furthermore, comparing the degradation rates of these single mutants to the triple mutant could clarify the redundancy and specificity of these motifs in the context of Nrm1 regulation.

We had done (but were not shown) the experiments requested by the reviewer. As the reviewer can observe, while in the full length (FL) Nrm1 the D-box seems to be contributing more than the KEN box or the Cbox in the context of the FL protein, the KEN box is more important in the context of the split protein (Nrm1 A). We have added this information to the Appendix PDF section of the manuscript.

11) While the manuscript references relevant literature, it would benefit from a more comprehensive discussion of how this study fits into the broader context of cell cycle regulation and transcriptional control. Emphasizing the novel aspects of this work in relation to previous studies would strengthen the manuscript.

We have included a couple of sentences at the end of the first paragraph of the Discussion, following the advice of the reviewer, which will help to position the results shown in this manuscript: *“Importantly, Nrm1 alone has minimal repressive activity; it primarily functions as a cell cycle–dependent sensor that facilitates the recruitment of Yox1 -the true repressor- to the MBF complex. This mechanism represents a significant difference from the regulatory systems described in budding yeast and higher eukaryotes.”*

Minor comments

1) The current manuscript title might be confusing. It is not immediately evident what the authors mean by the word "bistable" in the title.

We have to admit that this was not our original idea; a bistable mechanism has been previously proposed for mammalian cells as the mechanism regulating the G1/S transition (see for example Yao et al (2008), Nat Cell Biol 10, 476–482). We have now shown that a somehow similar mechanism is imposed in fission yeast, although with many differences; although the endpoint is the same.

2) The authors might want to rephrase the following sentence in the abstract: "This mechanism overlaps with the irreversible degradation of unphosphorylated Nrm1 (which originates from de novo synthesis or by dephosphorylation of pre-existing Nrm1 during anaphase), preventing its re-association with MBF until the end of the following S phase.", as the degradation of proteins is always irreversible.

Thanks! We have deleted the word irreversible from the sentence: *“This mechanism overlaps with the degradation of unphosphorylated Nrm1 (which originates from de novo synthesis or by dephosphorylation of pre-existing Nrm1 during anaphase) ...”*

3) The authors should explain in more detail how ChIP was carried out in the Methods section. Also, what is the "control" region used in the experiment shown in Fig EV4D?

The control is mtDNA. We have included the relevant information at the methods section.

4) Please write all centrifugations as rcf, not rpm to ensure reproducibility.

We have changed rpm to rcf

5) Please verify that the genome assembly versions used for RNA-seq data alignment is correct. Right now, the used name "Schizosaccharomyces pombe all chromosomes 20230315" does not seem to correspond to official assembly versions.

The reference used is an official version of *S. pombe* genome: it was downloaded from the fission yeast database, Pombase, in the following link:

<https://www.pombase.org/data/releases/>

The version used corresponds to the 2023-03-01 release that was downloaded on 2023-03-15; the downloaded database may contain minor modifications from the original release.

6) Please add the software name used to assign reads to features in RNA-seq experiment.

HTSeq was used for transcript quantification. We have now included this information in the Methods section.

7) Page 19 sentence "Cells were kept at 4°C un 50 mM NaCitrate pH7" - should read "in", not "un".

Thanks! We have corrected the typo

8) Page 21, title "Life-cell airyscan time-lapse experiments for cell cycle quantification" should possibly be "live-cell."

Thanks, we have corrected the text.

9) Figure 2E - some phosphorylation still remains after inhibiting CDK1. Thus, the claim of "abolishes phosphorylation" might need to be softened.

We use a bulky ATP to inhibit Cdc2-asM17, which does not mean that is 100% inhibited. We have changed the sentence to "*this phosphorylation-dependent shift was almost abolished in metaphase-arrested cells expressing a CDK1-analogue sensitive mutant (cdc2-asM17), only after the addition of 1-NM-PP1, implicating that CDK1 is directly or indirectly involved in Nrm1 phosphorylation*"

10) Figure 3A - the figure would benefit from adding the color code information as a legend in the figure itself. Also, depicting where the Cbox is located would be beneficial.

We have added the color code in the figure legend and described the Cbox.

11) The authors should clearly define what the dCB and dCpnt mutations are. Currently, the reader has to assume that the dCB mutation involves replacing all ten residues with alanine, while the dCpnt mutation involves changing five amino acids as indicated in the text. However, the text in the Results section (page 10) states: "To evaluate the contribution of these domains to Nrm1 stability, we introduced mutations in these motifs-either replacing all ten residues with alanines or introducing point mutations in two residues of each motif-and measured the half-life of the resulting mutants." This implies that the dCpnt mutation involves four mutations, not five, which needs clarification.

We have corrected the text. Basically, Δ Cbox corresponds to 10 alanine substitutions (positions 251-255 and 266-270); Δ Cpnt is just 5 of them: H252A W255A, R263A, V265A, L267A (in which, 2 correspond to each one of the domains and the fifth to the linker between them).

Dear Jose,

Thank you for submitting your revised manuscript. It has now been seen by two of the original referees. My apologies for the delay in getting back to you, which was due to recent conference travels.

As you will see, referees find that the study is significantly improved during revision and recommend publication. However, the editorial points below need to be addressed before I can accept the manuscript.

- Please address the remaining concern of referee #3.
- Please reduce the number of keywords to 5.
- Please make the dataset GSE287772 publicly available and remove the reviewers' token from the manuscript text.
- In line with the previous point, please provide a URL for GSE287772, which directly resolves to the dataset in the Data Availability section.
- Please remove the Author Contributions section from the manuscript text.
- As per our format requirements, in the reference list, citations should be listed in alphabetical order and then chronologically, with the authors' surnames and initials inverted; where there are more than 10 authors on a paper, 10 will be listed, followed by 'et al.'. Please see <https://www.embopress.org/page/journal/14693178/authorguide#referencesformat>
- We note that the Author Checklist is missing responses to cells D99-D101.
- The funding information needs to be complete in the manuscript text and the manuscript tracking system as per EMBO Press policy. We note that FI predoctoral fellowship from Generalitat de Catalunya has not been entered into the manuscript tracking system.
- We note the following regarding the figure callouts: Fig. S3B is a wrong nomenclature according to our format requirements and should be corrected. Please double check to ensure correct panel is called out.
- Along similar lines, Table S1 and Table S2 are also not correct callouts, which should be corrected as Table EV1 and EV2, respectively. Please remember to update all callouts, including in the Reagents & Tools table.
- Please add a Table of Contents to the Appendix file with page numbers on the title page.
- We note that source data for the Sty1 blot (uncropped version) of Figure 2E was not provided.
- Moreover, in the source data checklist, we note that you state that Fig. 4B and C are deposited into public repositories. Please add the accession numbers of these datasets into the source data checklist for clarity.
- We note that previously published MBF ChIP data was used for re-analysis in Aligianni et al, 2009; Skribbe et al, 2025. Please cite the specific datasets used in this study separately in the form of data citation. Please see <https://www.embopress.org/page/journal/14693178/authorguide#referencesformat> for examples of dataset citations.
- Main figure legends, tables and EV Figure legends should be placed after the References, at the end of the manuscript. Separate title page for EV legends should be removed.
- Our production/data editors have asked you to clarify several points in the figure legends - Figure Legends (main + EV):
 - o Please note that the exact p values are not provided in the legends of figures 1A, 3D, E; 5B, E, F, G; 6D, EV1 C, F, H; EV4 A, B, C; EV5 D
 - o Please indicate the statistical test used for data analysis in the legend of figure 6D
 - o Please note that the box plots need to be defined in terms of centre, percentile in the legend of figure 6D
- Papers published in EMBO Reports include a 'synopsis' and 'bullet points' to further enhance discoverability. Both are displayed on the html version of the paper and are freely accessible to all readers. The synopsis includes a short standfirst summarizing the study in 1 or 2 sentences (max 35 words) that summarize the paper and are provided by the authors and streamlined by the handling editor. I would therefore ask you to include your synopsis blurb and 3-5 bullet points listing the key experimental findings.

Thank you again for giving us to consider your manuscript for EMBO Reports, I look forward to your minor revision.

Kind regards,

Deniz

--

Deniz Senyilmaz Tiebe, PhD
Senior Scientific Editor
EMBO Reports

Referee #1:

The authors have satisfactorily addressed all of my concerns.

Referee #2:

The authors have addressed and satisfactorily answered all my comments. I agree with their evaluation that the manuscript has been further improved during revision. They provide a thorough analysis of the specifics of cell-cycle regulated transcription in fission yeast, consequences of losing this regulation, and discuss interesting similarities and differences to analogous regulation in budding yeast and metazoans.

Referee #3:

We thank the authors for their work in addressing our comments experimentally and agree with their assessment that the manuscript has been improved to meet the standards of EMBO Reports. We support its publication.

If there are space limitations, the AlphaFold section could be eliminated, as the authors themselves point out its speculative purpose: "We have included AlphaFold predictions not to infer possible interactions between different proteins, but to show that our experimental findings may correlate with possible structural predictions."

The authors addressed the remaining editorial issues.

Prof. José Ayté
Universitat Pompeu Fabra
Doctor Aiguader, 88
Barcelona 08003
Spain

Dear Jose,

Thank you for submitting your revised manuscript. I have now looked at everything and all is fine. Therefore, I am very pleased to accept your manuscript for publication in EMBO Reports.

Congratulations on a nice work!

Before we can export your manuscript to our production team, I need your input on one more point. I note that the below point raised by our data editors has not been addressed. I have attached the final version of the manuscript text. You can add the relevant information to the file and send it to me per email. Thank you.

"Please note that the exact p values are not provided in the legends of figures 1A, 3D, E; 5B, E, F, G; 6D, EV1 C, F, H; EV4 A, B, C; EV5 D"

Kind regards,

Deniz

--

Deniz Senyilmaz Tiebe, PhD
Senior Scientific Editor
EMBO Reports
